# Proactive and reactive accumulation-to-bound processes compete during perceptual decisions

Lluís Hernández-Navarro [1], Ainhoa Hermoso-Mendizabal [1], Daniel Duque [1], Jaime de la Rocha [1,4] & Alexandre Hyafil [2,3,4 ✉]

Standard models of perceptual decision-making postulate that a response is triggered in reaction to stimulus presentation when the accumulated stimulus evidence reaches a decision threshold. This framework excludes however the possibility that informed responses are generated proactively at a time independent of stimulus. Here, we find that, in a free reaction time auditory task in rats, reactive and proactive responses coexist, suggesting that choice selection and motor initiation, commonly viewed as serial processes, are decoupled in general. We capture this behavior by a novel model in which proactive and reactive responses are triggered whenever either of two competing processes, respectively Action Initiation or Evidence Accumulation, reaches a bound. In both types of response, the choice is ultimately informed by the Evidence Accumulation process. The Action Initiation process readily explains premature responses, contributes to urgency effects at long reaction times and mediates the slowing of the responses as animals get satiated and tired during sessions. Moreover, it successfully predicts reaction time distributions when the stimulus was either delayed, advanced or omitted. Overall, these results fundamentally extend standard models of evidence accumulation in decision making by showing that proactive and reactive processes compete for the generation of responses.

[1] IDIBAPS, Rosselló 149, 08036 Barcelona, Spain. [2] Center for Brain and Cognition, Universitat Pompeu Fabra, Ramón Trias Fargas, 25, 08018 Barcelona, Spain. [3] Center of Mathematical Research, Campus UAB Edifici C, 08193 Bellaterra (Barcelona), Spain. [4] These authors contributed equally: Jaime de la Rocha, Alexandre Hyafil. ✉email: alexandre.hyafil@gmail.com

O ur brains must constantly make perceptual decisions in the face of ambiguous and noisy stimuli. A very successful framework in cognitive psychology and neuroscience suggests that, under free-response conditions, these perceptual decisions follow an accumulation to bound policy: sensory evidence is integrated over time until a given decision criterion is met[1–3]. Computational models implementing this idea, such as the drift-diffusion model (DDM) and race models, provide a good quantitative account for reaction times (RTs) and choices in humans, non-human primates, and rodents[4–10]. A common assumption of the standard DDM and other accumulation models is that perceptual decision-making is essentially a serial process where the motor response is initiated reactively, only once the evidence accumulation has reached the decision boundary. Hence, choices and RTs are intrinsically coupled.

In real life, however, motor actions are not always triggered by a stimulus or external event. Proactive responses are at play in self-paced actions in the absence of stimuli[11–13], like in foraging decisions[14], but they can also be particularly prevalent in sensory-driven decisions when the stimulus onset can be anticipated[15] or under strong time pressure[16–19]. Proactive responses could also be a convenient strategy if the response can be updated after it is initiated to incorporate new sensory information. Moreover, even when subjects accumulate a fixed amount of sensory evidence before eliciting a response, reaching this criterion may at times take too long[20–22]. In those circumstances, subjects may need to proactively terminate the accumulation process and trigger a response independently of the accumulated evidence. Therefore, although previous studies characterizing proactive and reactive responses have implicitly assumed that they represent distinct response modes deployed in different contexts, the two may actually coexist. The possible coexistence of proactive and reactive processes remains unknown, and so are the mechanisms describing their possible interactions.

To address these questions, we analyzed the RTs and choices of rats in an auditory discrimination task[23]. Our results show that, in addition to reactive responses triggered by sensory evidence accumulation, animals exhibit informed proactive responses whose timing is independent of the stimulus. This coexistence was captured by a perceptual decision-making model where responses can be triggered by either of two parallel dynamical processes: Action Initiation, yielding proactive responses; or Evidence Accumulation, yielding reactive responses. The model correctly captured reaction times distributions and choices, and their dependence on both stimulus characteristics as well as internal variables such as satiety and fatigue. Overall, these results extend standard models of decision-making by showing that, in freely-timed paradigms, both evidence accumulation and action initiation run in parallel and compete with each other to be the triggers of our actions.

## Results

**Reaction time auditory task.** We trained rats in two variants of a reaction-time auditory discrimination task[23] (Fig. 1). On each variant, an acoustic stimulus was played after a 300 ms fixation period at the center port, and interrupted once the animal withdrew from the center port to make a response by poking at one of the two side ports. Since the fixation period was constant, animals could predict the time of the stimulus onset. Withdrawal of the center port before stimulus onset, i.e., fixation breaks (FB), led to trial abortion. The stimulus was a superposition of two amplitude-modulated (AM) sounds and the animals were rewarded for correctly discriminating the sound with the higher average intensity. The two sounds were either two pure tones with different frequency (frequency discrimination task, Group 1,

$n = 10$) or two noise bursts coming from the left and the right speaker (laterality discrimination task, for Groups 2–4, $n = 16$). The discrimination difficulty of each stimulus (i.e., stimulus strength $s$) set the relative amplitude of each sound. Stimulus sequences included serial correlations to study expectation-mediated choice biases (see Supp. Methods). For our current analysis, we focus however on trials following an error, where we have previously shown that animals do not leverage on the serial correlations to bias their choices (unbiased trials)[23] (Supplementary Fig. 1). The results obtained in this condition also held for expectation-biased trials (see below).

**Decoupling of reaction times and choices.** Standard models of evidence-to-bound integration predict that, as the stimulus strength increases, accuracy also increases, while reaction time (RT) decreases[8–10]. As expected, in the frequency discrimination task, RTs did indeed decrease with stimulus strength $s$, indicating that at least a fraction of responses were reactive, although the overall mean modulation was weak (~2% FOV explained). Surprisingly, however, RTs shorter than ~90 ms were independent of stimulus strength (Fig. 2a). To assess more precisely how stimulus strength modulates RTs, we computed time delay curves, which measured the advancement as stimulus strength increased of responses below a given RT (Fig. 2b; "Methods" and Supplementary Fig. 2)[24]. The time beyond which RTs showed a significant modulation with $s$ was consistent across rats (mean stimulus modulation onset M = 95 ms, SD = 20 ms; vertical line in Fig. 2b). We called *express responses* the

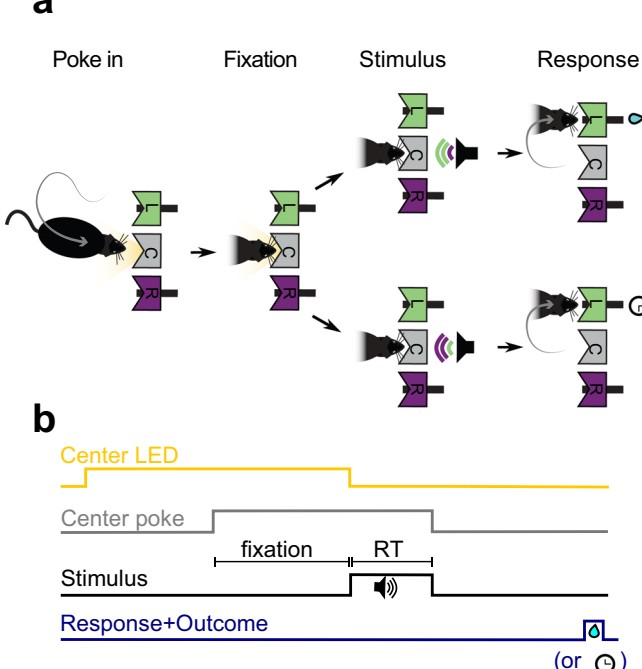

**a**

**b**

**Fig. 1 Auditory discrimination task. a** The initiation of each trial was cued by a center port LED (yellow). Following the cue, rats poked in the center port (C, gray) and the fixation period (300 ms) started. At the end of the fixation period, the stimulus was presented, consisting of a mixture of two AM sounds, each of which is associated with reward in the Left (L, green) or Right (R, purple) port. The two sounds differed in frequency (frequency discrimination task) or in the location of the sound source (lateralization discrimination task). After stimulus onset, the rats were free to withdraw from the center port to elicit a response, causing the stimulus to stop. Correct responses were rewarded with water and incorrect responses were punished by a time-out. **b** Task temporal scheme: reaction time (RT) denotes the time from stimulus onset to center poke withdrawal.

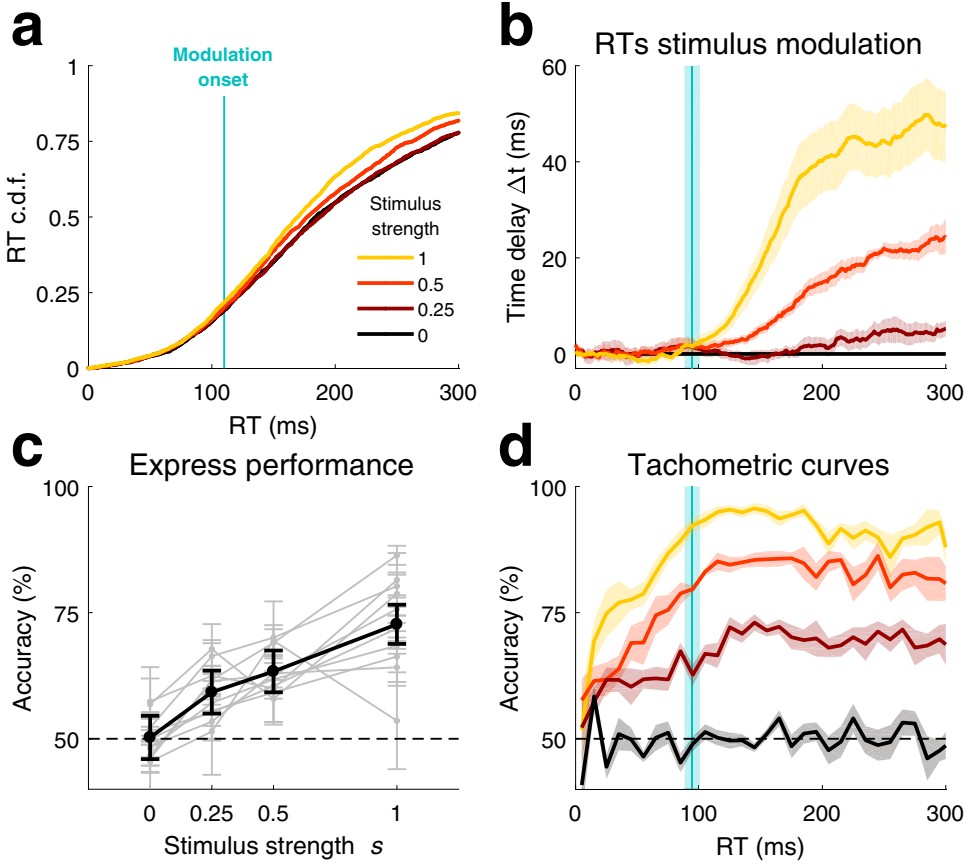

**Fig. 2 Decoupled reaction times and choices for express responses. a** RT cumulative distribution function (RT cdf) for different values of stimulus strength for an example animal (rat #11) in unbiased trials. Vertical line indicates the onset of the stimulus modulation on RT cdf computed for this rat (see "Methods"). **b** Mean time delay curves for RT cdf in the frequency discrimination task, averaged across rats (Group 1, $n = 10$). Trials with uninformative stimulus (stim. strength $s = 0$) are used as reference (i.e. time delay $\Delta t = 0$ for $s = 0$). Vertical line represents the mean modulation onset time (95 ms). Shaded areas: standard error of the mean (s.e.m). **c** Accuracy on trials with RTs shorter than 50 ms as a function of stimulus strength $s$. Thick line denotes the group average across the same rats as in panel (**b**); thin lines represent individual rats. Error bars represent s.e.m. **d** Tachometric curves, i.e., accuracy as a function of reaction times[16], for different stimulus strength $s$ values, averaged across rats. Reaction times were binned in windows of 10 ms. Vertical line: mean RT-stimulus modulation onset. Shaded areas: s.e.m. Source data are provided as a Source data file.

responses occurring between stimulus onset and this modulation onset, which accounted for 35% (SD = 25%) of all responses (Note that the express responses in our task are not equivalent to express saccades in oculomotor tasks[25]; see the "Discussion" section). We wondered whether express responses corresponded to rushed responses in which rats did not have time to process the stimulus, and made a guess choice based on stimulus-independent factors. We found, however, that categorization accuracy for express responses was well above chance and increased with stimulus strength in all animals, demonstrating that rats used the stimulus to determine their choice even in express responses (Fig. 2c). This dependence was well illustrated by the tachometric curve, representing how choice accuracy varies with RT (Fig. 2d)[16]. During the interval of express responses, accuracy increased, starting at RTs < 10 ms, until it reached a maximum and a relative plateau for non-express responses. These results also hold for both expectation-biased trials and for the laterality version of the task (Supplementary Figs. 3 and 10).

Overall, the timing of rat responses seemed to follow two modes: express responses, where the timing was independent of evidence accumulation but the choice did depend on the stimulus; and slower responses, where both the timing and the choice accuracy depend on the stimulus. Express responses are incompatible with standard models of evidence accumulation and their extensions because these models inherently rely on evidence

bounding to trigger the response, and therefore they invariably predict (1) that if the stimulus impacts the choice, it should also impact reaction times[20,21,26], and (2) that the stimulus can only impact the choice after the non-decision time, which includes sensory and motor delays. Slower responses in contrast, could in principle be triggered by the same evidence accumulation process giving rise to choice selection, and thus seem compatible with standard models of evidence accumulation. Because of this dichotomy, we next developed a general model for perceptual decision-making that could account for the full spectrum of rats' responses found in our task.

**A parallel model of action initiation and sensory evidence integration captures rats responses**. To capture express responses, we introduced the Parallel Sensory Integration and Action Model (PSIAM) composed of: (1) a standard Evidence Accumulation process (EA) that integrates stimulus evidence over time and that is bounded by left and right decision bounds, i.e., a standard DDM (Fig. 3b, top)[8–10]; (2) an independent Action Initiation process (AI) which reflects the preparation of a response. Because the fixation period preceding the stimulus was fixed, rats could prepare the motor action during this period in order to respond rapidly after stimulus onset, while maintaining a reasonable accuracy. The AI process represented this proactive timing signal[27], and was implemented as a single-bounded

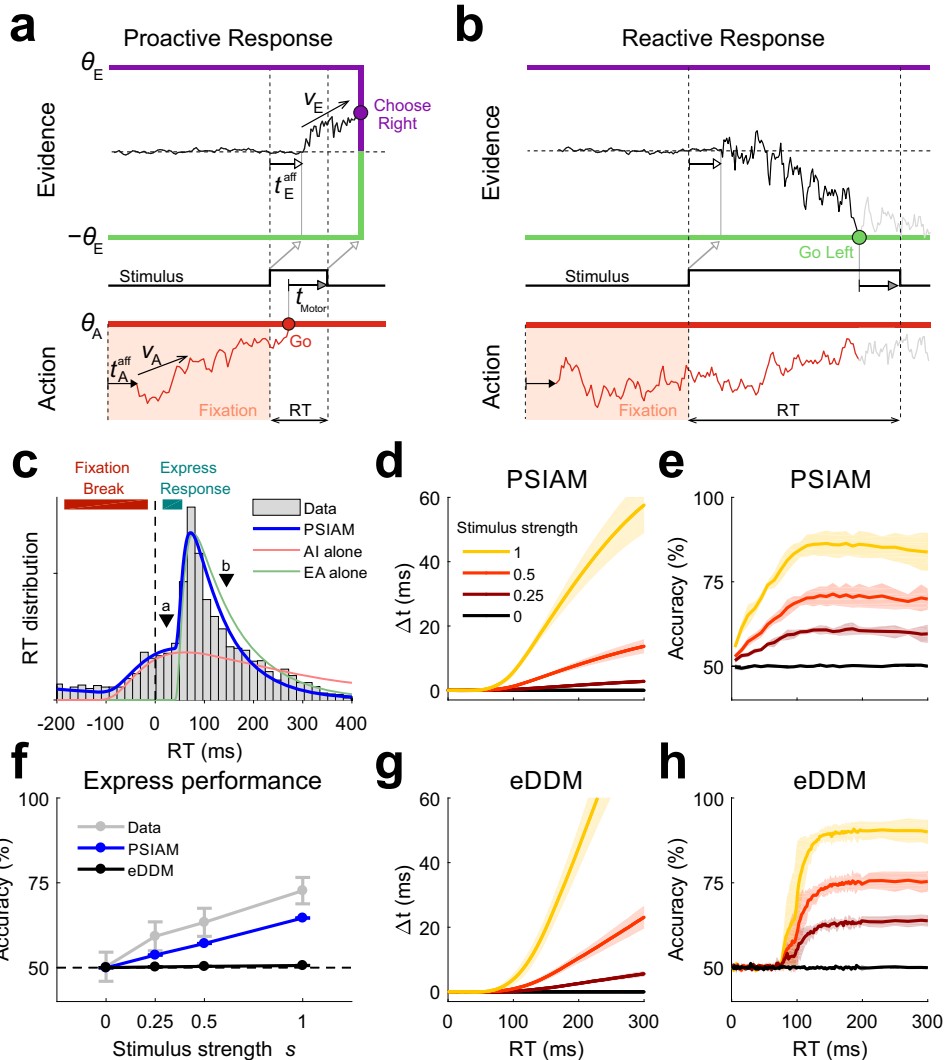

**Fig. 3 Parallel Sensory Integration and Action model. a** Example proactive response generated by the PSIAM. The action initiation process (AI, red trace) starts ramping up with afferent latency $t_A^{aff}$ after the fixation onset, with mean drift $\nu_A$, until it hits the Go bound $\theta_A$. The evidence accumulation process (EA, black trace) starts in response to the stimulus onset and after an afferent, sensory latency $t_E^{aff}$, with a stimulus-dependent mean drift $\nu_E$, and symmetric decision bounds $\pm\theta_E$. RT is determined by the first of the AI and EA processes reaching a bound. Here, AI reaches the Go bound first, triggering a proactive response after a motor latency $t_{motor}$. The stimulus is stopped at the response time (middle scheme). The choice is then determined by the final sign of the EA process, after the full stimulus has been integrated. This read-out is illustrated by an instantaneous collapse of the decision bounds. **b** Example reactive response, where the EA process reaches a decision bound first, setting both RT and choice; the AI process plays no role. **c** RT distribution for an example rat (gray bars; rat #12) and model fit (blue line) for maximum stimulus strength $s = 1$. The distributions of proactive (red) and reactive (green) RTs generated by the AI and EA processes working in isolation are also shown. Triangles correspond to example responses shown in panels **a-b**. **d** Simulated time delay curves, generated from the PSIAM fit to RTs for each animal, averaged across rats (Group 1, $n = 10$). Shaded areas: s.e.m. **e** Simulated tachometric curves, predicted from fitting the PSIAM to each animal RTs, averaged across rats. RT bin width: 10 ms; shaded areas: s.e.m. **f** Accuracy on express response trials with RTs shorter than 50 ms as a function of stimulus strength $s$ (gray), and for model predictions from the PSIAM (blue) and the extended DDM (black), both fitted to individual rat RTs. Dots show average across rats. Error bars: s.e.m. **g, h** Same as in panels **d, e** for the eDDM. Lines: median across rats. Shaded areas: median absolute deviation. Source data are provided as a Source data file.

diffusive process that ramped until it reached the Go bound, independently of stimulus. The PSIAM can generate two types of responses, proactive or reactive, depending on which of the two processes, the AI or the EA, reaches a bound first. If the AI process hits the Go bound first (proactive response; Fig. 3a, bottom) a response is triggered, and the choice is given by the sign of the final accumulated evidence, i.e., the sign of the EA process after the stimulus, which is interrupted when the rat withdraws from the center port, is fully integrated (Fig. 3a, top). This procedure is equivalent to an instantaneous collapse of EA bounds (Fig. 3a). In proactive responses, the timing of the

response is thus completely independent of the stimulus strength. Conversely, if EA reaches a decision bound first (reactive response), it sets both RT and choice, and AI plays no role, as in a standard DDM (Fig. 3b). In this framework, express responses correspond to proactive responses that took place before the EA could start triggering any response: this time corresponds to the sum of EA latency $t_E$ and the minimal EA integration-to-bound time (Fig. 3a).

We assessed the capacity of the PSIAM to quantitatively capture rat behavior by fitting the model to the RTs. The model nicely captured the shape of the RT distributions for each animal

by identifying the two underlying components, i.e., the proactive and reactive responses (Fig. 3c; Supplementary Fig. 4). This separation allowed the model to readily account for the fixation breaks (FBs), interpreted as proactive responses occurring before stimulus onset (Fig. 3c). The model also captured how the upward deflection in the RT distributions, reflecting the earliest reactive responses, increased with stimulus strength (Supplementary Fig. 4, last column). The agreement between data and model in the RT pdfs is also visible in the time delay curves, which, as previously shown for the data (Fig. 2b), were all collapsed at zero for express RTs, and displayed increasing modulation by stimulus strength afterward (Fig. 3d). Furthermore, the model also captured the shape of the experimental tachometric curves (Fig. 3e). Note that animal choices were not used to estimate the parameters of the model, so the tachometric curves are direct predictions from the model (see "Methods"). The model captured the increase in accuracy for fast RTs (RT < 100 ms), followed by a relatively flat plateau for intermediate RTs (from 100 to 300 ms). More importantly, the model provided an explanation for the shape of the tachometric curves: for express responses, where the time of integration of the EA process is set by the AI process, longer RTs allow for longer integration of stimulus evidence, leading to higher accuracy. Intermediate RTs were dominated on the other hand by reactive responses caused by the EA reaching a decision bound. For such responses, the PSIAM is equivalent to the standard DDM with constant bounds, where the accuracy is independent of RT[8–10]. This accounts for the relatively flat tachometric curves at each stimulus strength $s$ for these intermediate RTs. Of note, the PSIAM also captured the RT distributions and reproduced the time delay and tachometric curves for biased trials (Supplementary Figs. 5 and 6a-b). Overall, the PSIAM successfully captured rats' complex patterns of RTs and choices by stipulating that response times were controlled by parallel, independent evidence accumulation and action initiation processes.

The existence of stimulus-informed express responses is incompatible with a decision mechanism where the timing of the response relies only on the accumulated evidence reaching a decision boundary. Although such a mechanism can generate noise-triggered express responses, choices are only influenced by the stimulus for RTs larger than the non-decision time, which is typically at least 50 ms. We confirmed this by fitting an extended DDM (eDDM), featuring integration of internal noise prior to stimulus onset[28], to individual rat RTs. Pre-stimulus noise integration produced anticipatory responses as those seen in rats, although a formal model comparison showed that the PSIAM captured the overall RT distribution much better ($\Delta$BIC > 100 for any rat in Groups 1 and 2, $n = 16$; Supplementary Figs. 7j-k and 12a; see Supp. Methods). Crucially, the eDDM consistently predicted chance performance for express responses (< 50 ms, Fig. 3f–h, Supplementary Fig. 7f), unlike what we observed in animals. The limitation to generate informed choices earlier than sensorimotor delays is general to the joint RT-and-choice decision mechanism: it also applies if the DDM bounds change in time[20] or in the accelerated race-to-threshold model[16], where the stimulus modulates a race-to-threshold between anticipatory signals (Supplementary Fig. 7g-h,l; see Supp. Methods). In conclusion, the influence of stimulus on choice in express responses was very well captured by the PSIAM but was fundamentally at odds with standard models of decision-making relying on a single boundary mechanism for response initiation and choice selection.

**Proactive responses prevent longer reaction times**. While our AI process can be thought of as an independent urgency signal,

urgency signals have been mostly hypothesized to play a role in avoiding very long reaction times, rather than promoting very fast responses[20–22]. We thus wondered whether the AI could play a role beyond express responses, at long RTs. By construction, PSIAM responses are faster when both EA and AI processes are taken into account than when EA is considered in isolation (Fig. 4a, blue and green curves, respectively; Supplementary Fig. 8). Proactive responses dominated fast RTs, whereas reactive responses took over for intermediate RTs; however, proactive responses contributed as much as reactive responses at long RTs (Fig. 4b; Supplementary Fig. 9). These long proactive responses imply that, in those trials, the EA process had still not reached the decision bound, and thus the AI process acted as an urgency signal that prevented very slow reactive responses. Moreover, because in those trials the evidence level is lower than the decision bound, their associated accuracy should also be lower than for reactive choices with the same RT. Contaminant responses, included in the model as rare responses that occur at an approximately constant rate independently of the AI or EA processes, also increase in relative frequency at long RTs (black lines in Fig. 4a, b; Supplementary Figs. 8 and 9; see "Methods"); and because they yield random choices, their occurrence also predicts lower accuracy. Hence, the increase in the fraction of both proactive and contaminant responses should thus be accompanied by a reduction in categorization accuracy. We tested this prediction by examining tachometric curves at these very long, infrequent RTs and found a significant decrease in accuracy for long RTs (Fig. 4c, d). Although contaminants alone could also produce a decay of the tachometric curves at long RTs, proactive responses in the PSIAM contributed significantly to accentuate and advance this decrease (Fig. 4d). In sum, AI did not only promote express responses but it also shaped slow responses, hindering their occurrence at the expense of reduced accuracy for long RTs.

**Manipulating stimulus presentation uncovers proactive and reactive responses**. The PSIAM core hypothesis is that, on each trial, animals prepare a proactive response which is initiated independently of the presentation of the stimulus, unless a reactive response is triggered first. The model then predicts that, if the stimulus is removed, animal responses should all be proactive responses triggered by the AI process. We tested this in a second group of rats (Group 2, $n = 6$; laterality discrimination task) where we omitted the presentation of the stimulus in a subset of trials but still rewarded one of the two responses (silent catch trials, 10%). In such trials, animals made choices using internal estimates instead of the stimuli. The stimulus was normally played in the rest of the trials of the session (standard trials, 90%) (Fig. 5a, middle and top). The RT distributions for standard trials were very similar to those observed in Group 1 (Fig. 5b; Supplementary Figs. 11 and 12a). Crucially, the distribution of RTs in silent trials was accurately predicted by the distribution of proactive responses alone obtained from fitting the full PSIAM in standard trials (Fig. 5c; Supplementary Figs. 11 and 12b). By contrast, the extended DDM failed to capture the distribution of RTs in silent trials ($\Delta$LLH < −25 for all rats; Supplementary Fig. 12b; see Supplementary Methods). Hence, removing the stimulus fully unveiled the RT distribution of proactive responses that was mixed with RTs from reactive responses in standard trials, providing evidence that within each trial the AI process evolves independently of the presentation of the stimulus.

The second basic hypothesis of the PSIAM is that proactive responses are triggered by a process initiated with fixation onset, while reactive responses are triggered by a process locked to stimulus onset. The model predicts that varying the interval

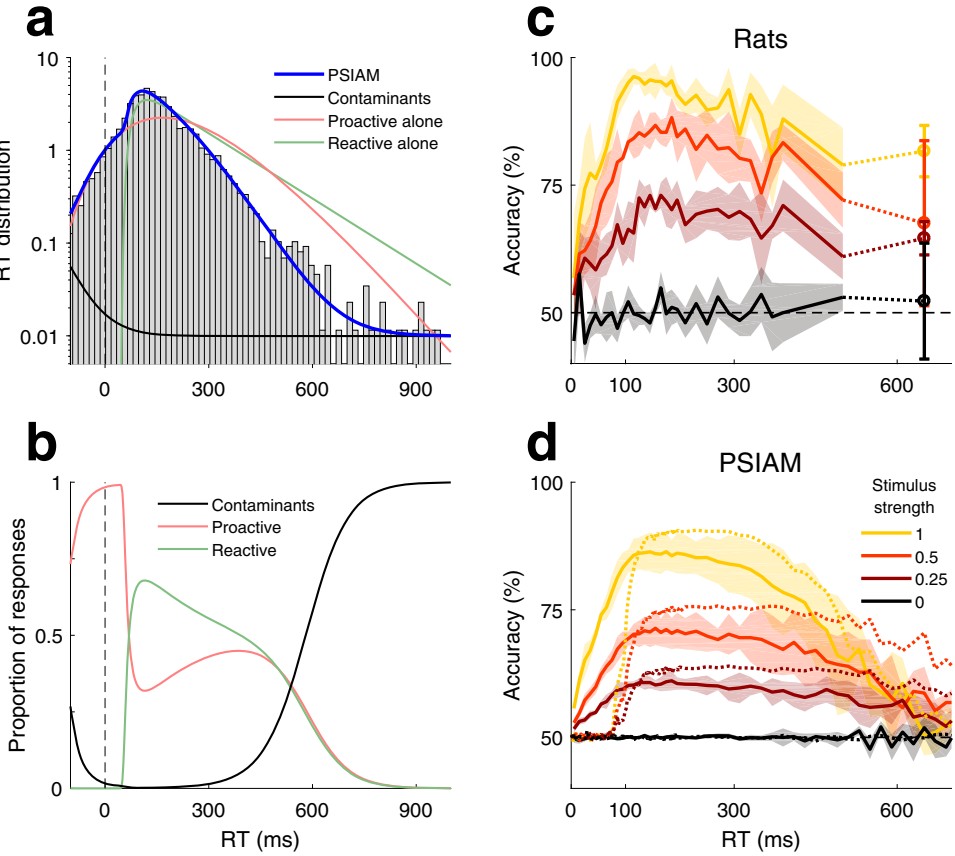

**Fig. 4 Reaction times and accuracy for long responses. a** RT distribution for an example animal (gray bars; rat #15) and model fit (PSIAM, blue line) for medium stimulus strength ($s = 0.5$). To better resolve the tail of the distribution we used a logarithmic scale. Notice how the tail of the PSIAM distribution has been substantially trimmed with respect to the distributions of proactive alone (red) and reactive alone (green) RTs generated by the AI and EA processes working in isolation. For very long RTs (above 600 ms), contaminant responses, modeled as random guesses, dominate the pdf (black curve; average $7 \pm 3\%$ of the total responses; see "Methods" and Supplementary Fig. 8). Distribution of contaminant RTs is shown in black. **b** Proportion of proactive (red), reactive (green) and contaminant (black) responses vs. RT for the same example rat as in panel (**a**). **c** Tachometric curves for different stimulus strength $s$ values, median across rats. Open circles indicate accuracy for responses of RT > 600 ms. Shaded areas and error bars: median absolute deviation. **d** Simulated PSIAM tachometric curves up to 700 ms, median across rats; shaded areas: median absolute deviation. Simulations of the eDDM are also shown (dotted lines) to illustrate the decay in accuracy caused by contaminant responses only. Source data are provided as a Source data file.

between fixation onset and stimulus onset should affect response timings by favoring one process over the other. In a second set of experiments, the stimulus was either advanced (advanced stimulus trials, $\Delta = -250$ or $-150$ ms for Groups 2 or 3, respectively; 5% of trials), or delayed (delayed stimulus trials, $\Delta = +50$ or $+150$ ms, Groups 2 or 3; 5% of trials) with respect to the standard condition ($\Delta = 0$, standard trials, 90% of trials) (Fig. 5a, bottom). We tested whether, in advanced and delayed stimulus trials, the RT distributions could be simply derived from the model fitted to standard trials but only shifting the onset of the EA process by the same amount $\Delta$. For delayed stimulus trials, the model with a shifted EA captured the RT distribution well: most responses were proactive, and the late stimulus onset caused a small bump of reactive responses at the tail of the RT distribution (Fig. 5e; Supplementary Figs. 13 and 14, right columns). This small impact of delayed stimulus was clearly visualized in the experimental time delay curves which showed that the RTs in delayed stimulus trials were significantly advanced with respect to silent trials as predicted by shifting the time delay curve of standard trials by 150 ms (Fig. 5f; $p$-value = 0.007 for delayed vs silent time delay curves at RT = 300 ms, one-sided $t$-test, Group 2). In contrast, for advanced stimulus trials, simply advancing the EA by $\Delta$ predicted faster responses than those observed in the rats (Fig. 5d, solid blue curve; Supplementary

Figs. 13 and 14, left columns). This suggests that, when stimulus is presented ahead of the expected time, the EA process takes longer to reach the decision bounds than in standard trials. Indeed, simply allowing the latency $t_E$ to be longer in these advanced trials provided a much better fit of the RT distribution (Fig. 5d, dashed blue curve; Supplementary Figs. 13 and 14, left columns). This additional delay, which increased with the magnitude of the advancement $\Delta$ (Fig. 5g), could reflect attentional delays related to the unexpected timing of the stimulus onset[29,30], or a signature of more complex interactions between AI and EA (see "Discussion"). Finally, adjusting the EA's drift $v_E$ instead of $t_E$ did not provide a better fit, and adjusting the two parameters together only caused a marginal average improvement (Fig. 5h; Supplementary Fig. 15). Overall, these experiments revealed that the PSIAM could account for how RT distributions are affected by changes in stimulus timing, and showed that EA latency is increased when stimulus timing is unexpected.

**Slowing of the responses along the session is mediated by a decrease in AI speed**. The existence of the dual mechanism for response timing in the PSIAM offers two possible mechanisms to account for changes in response speed: by a change in the

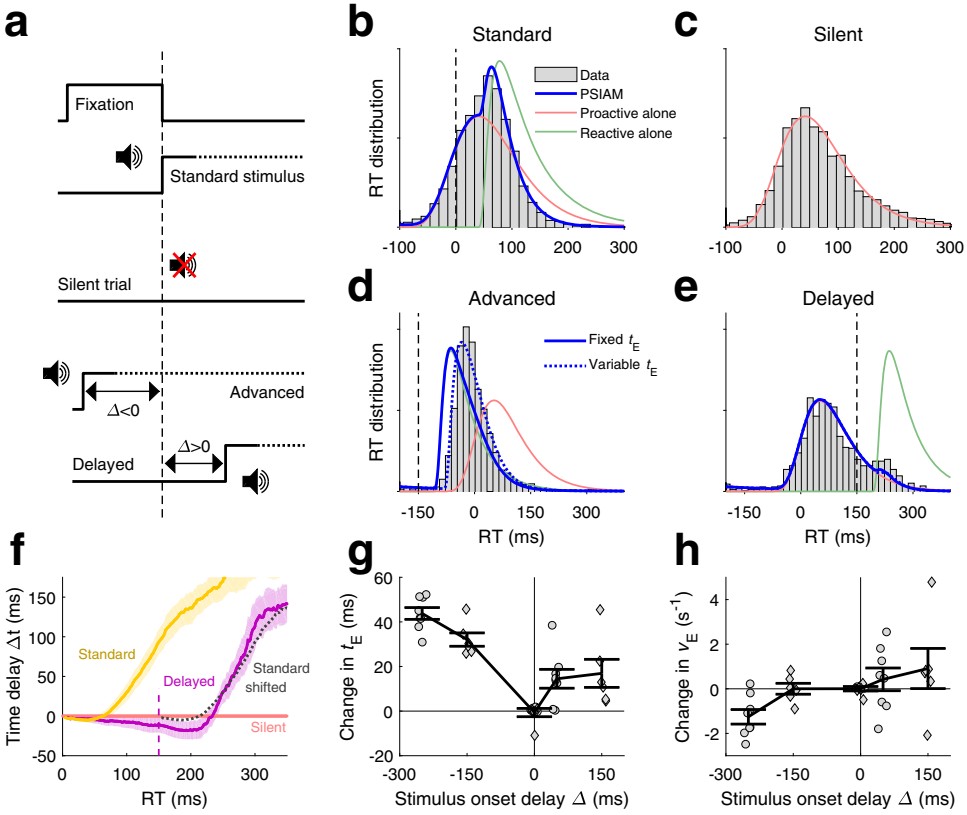

**Fig. 5 Reaction times in experiments with perturbed stimulus onset. a** In silent sessions, the stimulus was absent in 10% of trials (silent trial; Group 2, $n = 6$). In advanced/delayed stimulus sessions, the stimulus was advanced in 5% of trials ($\Delta = -150$ ms for Group 2 and $-250$ ms for Group 3, $n = 8$) and delayed in other 5% of trials ($\Delta = +150$ ms in Group 2 and $+50$ ms in Group 3) with respect to the standard onset time. **b** RT distribution for an example rat (gray bars; rat #43) and model fit (PSIAM, blue line) for maximum stimulus strength ($s = 1$) in standard trials in silent sessions. Red/green lines: RT distributions generated by the AI/EA processes in isolation. Vertical dashed line: stimulus onset. **c** RT distribution in silent trials from the same rat (gray bars). Model prediction corresponds to the RT distribution of proactive alone responses obtained from standard trials (red line; same as in **b**). **d** Same as in panel b but for a different example rat in advanced trials ($\Delta = -150$ ms; rat #44). Model prediction corresponds to advancing the reactive alone density from standard trials by 150 ms (green line) while keeping AI density (red line) unchanged (solid blue curve). Dashed blue curve shows model fit, where the EA latency parameter $t_E$ was fitted while keeping other parameters unchanged. **e** Same as in panel d for delayed stimulus trials, but delaying the isolated EA density by 150 ms in the model prediction. **f** Time delay curves in standard (yellow) and delayed ($\Delta = +150$ ms, purple) trials for maximum stimulus strength ($s = 1$) versus silent trials (reference, pink), averaged across rats. Dotted black line represents the standard condition shifted $+150$ ms for comparison. Vertical line: onset of the delayed stimulus. Shaded areas: s.e.m. **g**, **h** Difference in the fitted values for $t_E$ (**g**) and $v_E$ (**f**) at each stimulus onset delay $\Delta$. Line: average across animals; error bars: s.e.m.; diamonds (Group 2, $n = 6$) and circles (Group 3, $n = 8$): individual rats. Source data are provided as a Source data file.

decision bounds of the EA process, as classically reported[31]; or by a modulation of the speed of the AI process, taking place independently of evidence accumulation. We assessed whether the systematic slowing of the RTs observed within each session in almost every animal (Fig. 6a) was mediated by the modulation of AI, at odds with the classical account of slowing based on changes in the evidence accumulation process. Cumulative RT distributions sorted by trial index showed that there were fewer fixation breaks and fewer express responses at the end compared with the beginning of the session (Fig. 6b). In other words, the slowing already occurred for responses triggered even before any reactive response could be performed (Fig. 6c). This suggested that the slowing was due at least in part to changes in the AI. We thus incorporated within-session slowing in the model by allowing the drift of the AI to decrease linearly with trial number within each session, while keeping the EA parameters constant across the session. Remarkably, this simple modification captured quantitatively the change in time delay curves observed within the session (Fig. 6d). The model reproduced the non-monotonicity of the curves with a dip around 130 ms, indicating that the slowing in RTs was not as severe in that range of reaction times. In the

PSIAM, this corresponded to the range where reactive responses, which remained unchanged throughout the sessions, dominated over proactive responses (Fig. 6d, top inset; Supplementary Fig. 9). Overall, the analysis suggests that the AI process, but not the EA, was affected by internal factors that changed consistently within each session, mediating a gradual slowing of responses.

**Fatigue and satiety cause within session slowing by modulating AI speed.** The observed slowing of responses along the session could be linked to a change in two distinct internal variables of the animal[32-35] that increased as the session progressed: physical fatigue and satiety (Fig. 7a, center and right). The number of completed trials and the amount of consumed reward since the beginning of the session provided proxies for fatigue and satiety, respectively. In order to disentangle the contribution of each of these factors, we performed additional experiments where the size of the reward was varied across sessions (Group 4, $n = 4$; laterality discrimination task; see "Methods"). The proportion of fast proactive responses (RT < 50 ms) steadily decreased with trial index for all reward conditions, but the impact of reward size on RTs switched gradually along the session: while at the start of the

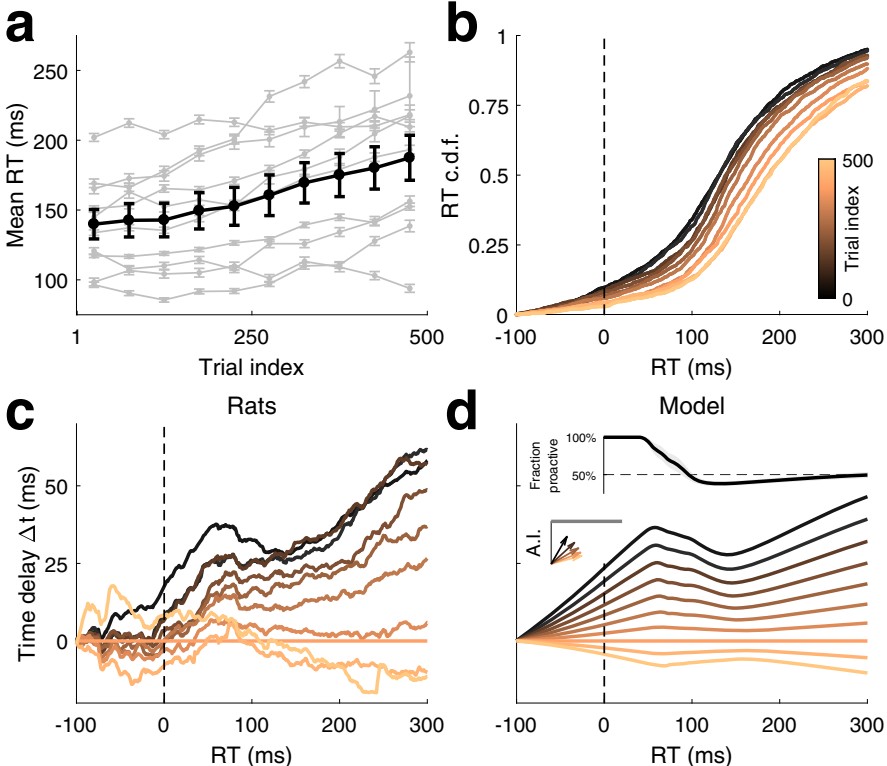

**Fig. 6 Within session slowing of responses. a** Mean RT as a function of trial position within the session. Thick black line: group average (Group 1, $n = 10$), thin gray lines: individual rats. Trials were grouped in 50 trial blocks. Error bars: s.e.m. **b** RT cumulative distribution for different trial blocks for an example animal (rat #14). Color code in panels (**b**–**d**) indicates trial block. Vertical dashed line in (**b**–**d**) indicates stimulus onset. **c** Time delay curves averaged across rats. Trial block 351–400 was taken as reference (i.e., corresponded to time delay $\Delta t = 0$). **d** Time delay curves simulated using the PSIAM fitted to the RTs for each animal, averaged across rats. Top inset: proportion of proactive vs reactive responses with RT as estimated by PSIAM for each animal, averaged across rats; shaded area: s.e.m. Left inset: sketch of the impact of trial index on AI's drift. Source data are provided as a Source data file.

session, a larger reward lead to a larger fraction of fast responses, the effect vanished at the end of the session (Fig. 7b). To explain this, we incorporated fatigue and satiety in the model by allowing the drift of the AI process to change linearly with trial number and with the amount of consumed reward within the session (Fig. 7a, top row). Moreover, since the reward size, fixed within a session, could also affect the rats' overall motivation and speed, we also let AI drift depend linearly with the reward size (Fig. 7a, left column). This extended model captured the impact of reward size on how the proportion of fast responses changed throughout the session (Fig. 7c). Trial index and consumed reward were found to have a negative impact on AI drift, i.e., slowed the AI process, whereas larger reward size had a positive impact, i.e., sped up the AI process (Fig. 7d). Thus, the gradual decay of the reward size dependence along the session arose from the opposite contributions of reward size and reward consumption: while at the start of the session, animals in large reward sessions were faster because they were more motivated, at the end of the session the effect vanished because they were also more satiated. In sum, the analysis shows that the AI process was subject to systematic variations within and across sessions by means of changes in three factors: while larger reward sped up the AI process, fatigue and satiety slowed it down.

## Discussion
We used an acoustic discrimination task to characterize the mechanisms underlying the timing and choices of rats' perceptual decisions. We found that the pattern of responses could not be solely explained by traditional models of perceptual decision

making whereby responses are triggered in reaction to the accumulated evidence crossing the bound[20,25,36]. In addition to such reactive responses, animals generated proactive responses in which the timing was stimulus-independent but the choice was stimulus-dependent (Fig. 2). We developed the PSIAM, a model proposing that each response type is triggered by one of two parallel processes: the Action Initiation (AI), a proactive timing process that starts in anticipation of stimulus onset and ramps up until reaching a Go bound; and the Evidence Accumulation (EA), a process initiated by the stimulus presentation that integrates sensory evidence until reaching one of the decision bounds (Fig. 3). The two processes run in parallel and race to induce a response, which is triggered as soon as one of the two reaches its respective bound. The Action Initiation process not only mediated premature responses but also generated slow responses, effectively implementing an urgency to respond when the EA process took too long to hit a decision bound (Fig. 4). As predicted by the model, proactive and reactive responses, which are locked respectively on the fixation onset and the stimulus onset, were dissociated in a second set of experiments where stimulus onset was perturbed (Fig. 5).

The express responses exhibited by our rats should not be confused with the so-called express saccades[25]. Express saccades occur at least 80 ms after stimulus onset, around 50 ms before the standard saccades, in conditions when subjects are pre-cued about the onset of the target stimulus (e.g., the gap paradigm)[37]. Despite strong debate about their underlying nature[37–39] it is generally accepted that they are triggered by the onset of the target stimulus[38] and hence they are reactive responses. The express responses made by our rats are what in psychophysics is

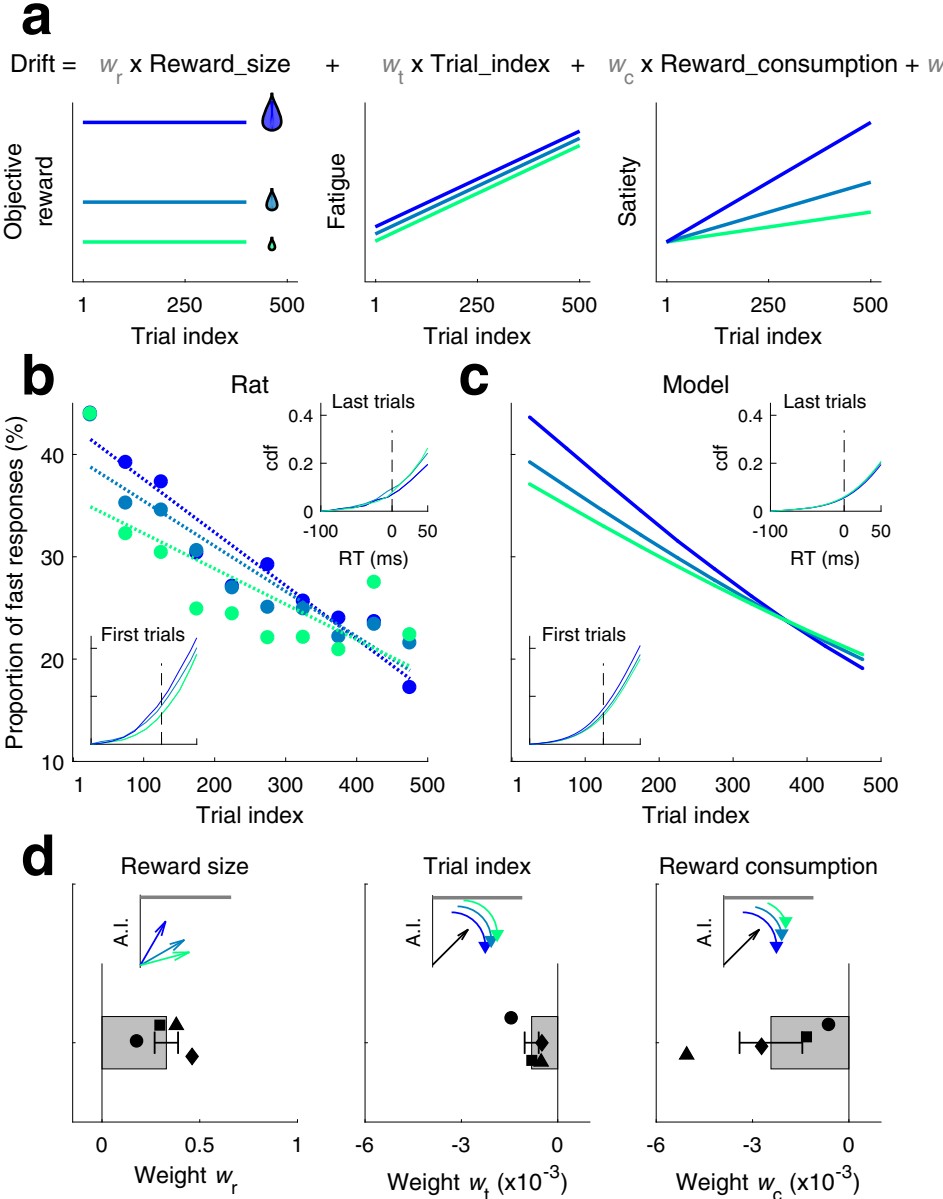

**Fig. 7 Internal factors impacting RTs. a** Reward size (left), fatigue (center), and satiety (right) as a function of trial position within session for low (green), middle (light blue), and high (strong blue) reward size sessions. Color code is shared for all panels (**a–d**). Top equation illustrates the implementation in the model of these three features as linear regressors in AI drift, with $w_r$ the weight for reward size, $w_t$ for fatigue, $w_c$ for satiety, and $w_0$ for the intercept. **b** Proportion of rat's fast responses (faster than 50 ms after stimulus onset, or fixation break) as a function of trial position within session and reward size, for an example animal (rat #54). Dashed lines: linear fits. Insets: RT cumulative distribution for trial block 51–100 (bottom left) and 451–500 (top right); vertical dashed line: stimulus onset. **c** Proportion of fast responses simulated using PSIAM fit to the RTs of the example rat in panel (**b**). Insets: same as in (**b**), but simulated using PSIAM. **d** Estimated weight for reward size (left), trial index (center), and reward consumption (right) regressors in AI drift. Bars: mean across animals (Group 4, $n = 4$); symbols: individual rats; error bars: s.e.m. Source data are provided as a Source data file.

called anticipatory responses, i.e., responses which are too fast to be triggered by the stimulus (see e.g., ref. [28]). They are usually considered as random responses and removed from subsequent analysis. Here we showed compelling evidence that, in our task, those anticipatory responses represented the lower tail of a wide distribution of proactive responses, whose timing was independent of the stimulus but whose choices were not random. For intermediate RTs, these proactive responses were present but they were mingled with reactive responses (Fig. 3c), as confirmed by experiments in which we omitted or shifted stimulus onset (Fig. 5).

Three factors in our task could be promoting the anticipation of the Action Initiation process leading to a large fraction of proactive responses. First, the stimulus onset occurred at a fixed time interval after fixation onset (i.e., a fixed foreperiod), so that the rats could reliably time their withdrawal from the central port right after stimulus onset[15]. A random fixation period[40] would probably lead to a more conservative strategy where the proactive process is slowed to avoid too early responses. Of note, temporal expectations also led to a reduction of the non-decision time of the EA process when the stimulus appeared at the predicted time in comparison to when it appeared earlier or later (Fig. 5g), in

agreement with a recent observation[41]. Second, because of sensory delays, and because the response involves an orienting body movement lasting around 300 ms, the rat had time for finishing stimulus integration between leaving the central port (which interrupts the stimulus) and poking at one of the lateral ports. This makes express responses a viable strategy in which subjects leave the port right after stimulus onset, before stimulus can influence the motor plan, and integrate the sensory information while executing the orienting movement. Preliminary analysis of the orienting trajectories reveals that the integration of the stimulus information is extremely fast, with the trajectories being updated in less than 90 ms after stimulus onset. The same type of online updating strategy could be applied for responses performed by arm or body movements, which can be swiftly adjusted en route to the target point[42]; but perhaps not for more ballistic responses such as saccades or button presses (but see ref. [43]). And third, the structure of the trial sequence allowed rats to partially predict the category of the upcoming stimulus based on previous trials' responses and rewards[23]. This partial predictability allowed animals to reach ~65% accuracy for minimal stimulus durations in biased trials (Supplementary Fig. 3; note however that most of our analyses were performed on unbiased analyses, where rats did not use trial history to predict the next stimulus category). These features allow achieving both very fast and accurate responses, especially for high stimulus strength where short stimuli provide enough evidence to reach high levels of accuracy[16] (Fig. 2d). In other words, animals are fooling the speed-accuracy trade-off[44].

The AI process allows modulating RTs not only based on the structure of the task, but also depending on motivational factors. We found that RTs slowed down within the session in all our animals (Fig. 6)[45]. Previous literature has investigated whether the impact of session time on RTs reflects the accumulated fatigue (physical and/or cognitive) or the increase of satiety. One study found that, in a delay discounting task, the increase in uncompleted trials observed during the session was larger for aged rats compared to young animals[32]. Since food consumption did not differ between the two age groups, the authors argued that the main driving factor was fatigue rather than satiation. In humans too, responses slow down when the subject is fatigued[46,47]. On the other hand, other studies have found that food or water satiety increases the latency of responses in rats and pigeons[33,34]. We disentangled the effect of fatigue and satiety by varying the reward size across sessions in additional experiments, and found that both factors contributed to within-session slowing (Fig. 7). Furthermore, our model-based approach allowed us to identify one possible cognitive mechanism underlying this effect: both fatigue and satiety impact the drift of the AI process, i.e., slow down the initiation of proactive responses. On the contrary, increasing reward size led to higher AI drift, and thus faster responses. This shows that internal states can modulate decision-making through the AI, without modulating Evidence Accumulation.

The idea that the timing of perceptual decisions depends on an internal sense of time pressure is not novel[20–22,26,48–50]. Several studies have proposed the existence of an urgency signal that interacts with the integration of sensory evidence in order to avoid slow responses. A popular idea is to incorporate urgency directly into the DDM, while keeping decision-making as a serial process where action initiation follows evidence integration. This can be done by gradually collapsing the decision bounds[20,26], or by a ramping evidence-independent signal that either increases the gain of evidence accumulation[21,50] or adds a non-specific input to all possible response activations, promoting motor actions[22,48,49,51,52]. All these mechanisms account for the fact that very slow responses are usually less frequent in participants than predicted by the DDM with fixed bounds. However, in all these extensions of the DDM, RTs always depend on the evidence accumulation since all responses are triggered after reaching a decision bound, so these models cannot explain express responses, where RT is independent of stimulus strength whereas choice is contingent on stimuli (35% of all responses in our dataset, SD = 25%; Fig. 2). Extensions to the DDM where the integration process starts prior to stimulus onset can produce anticipatory responses (Supplementary Fig. 7j). However, they inherently predict that if the RT is faster than the non-decision time, and thus independent of the stimulus evidence, then choice should also be independent of the stimuli (Fig. 3f–h and Supplementary Fig. 7e, f).

By contrast, the PSIAM provides an alternative approach where a single mechanism, i.e., proactive responses, can (1) account for express responses yielding stimulus-independent RTs and stimulus-dependent choices (Fig. 3d, e), and (2) contribute to the decrease of the probability of very slow responses (Fig. 4a). The PSIAM captured rat RT distributions better than an extended DDM (Supplementary Figs. 9j-k and 12a), and it parsimoniously predicted rat RTs in silent trials with remarkable accuracy (Fig. 5c and Supplementary Figs. 11 and 12b). In the PSIAM, AI sets a stimulus-independent internal deadline for triggering the response, avoiding long accumulation-to-bound processes in EA. As such, the action of the AI can be viewed as provoking the instantaneous collapse of decision bounds on the EA at a stochastic time defined by the AI bound crossing. While in our task the AI most often reached the Go bound shortly after stimulus onset, under other circumstances, a slower AI drift could be used to trigger proactive responses after a longer stimulus duration. Indeed, a slower and more precise AI process can shift the timing of proactive responses, from early to late responses, i.e., from causing express responses to ensuring an upper bound on integration time (Fig. 8a). Such a change also leads to a larger drop in accuracy associated with long RTs (Fig. 8b), an effect that is commonly observed in the decay of tachometric curves[51,53], but is usually attributed to either the variability in the drift of the evidence accumulation process or the gradual collapse of the decision bounds. Adjusting AI parameters in the PSIAM provided a more flexible way to set the speed-accuracy trade-off compared to modulating the decision bounds of the EA process[54,55]. Hence, while rats may lay on the impatient side of AI, AI in humans and non-humans primates performing classical perceptual tasks may primarily be used to prevent very long reactive responses[51]. Indeed, a model very similar to PSIAM (featuring a race between a timing process and an evidence accumulation process) was recently and independently proposed, providing a better account of human behavior than generalized DDMs in a variety of decision-making tasks[56]. This convergence of findings between species provided compelling evidence overall in favor of parallel proactive and reactive processes in the mammal brain.

The PSIAM is also related to previous models of sensory guided responses with a proactive component. In a classic mechanistic model of saccade initiation, Trappenberg and colleagues presented a circuit model of the superior colliculus where exogenous visual signals and endogenous preparatory signals are combined to generate target-directed saccades with variable RTs[39]. The model elegantly reproduces the impact of distractors and target location biases on saccade RTs, and generates express saccades when the target onset is cued (see above). However, despite the parallelism between proactive/reactive and endogenous/exogenous processes, this model exclusively generates either reactive responses or proactive random guesses. Reactive responses' latency can indeed be modulated by endogenous signals, but they are ultimately triggered by the visual input. In a different study, human subjects were forced to respond "earlier-than-normal" in a visually guided reaching task[18]. A simple

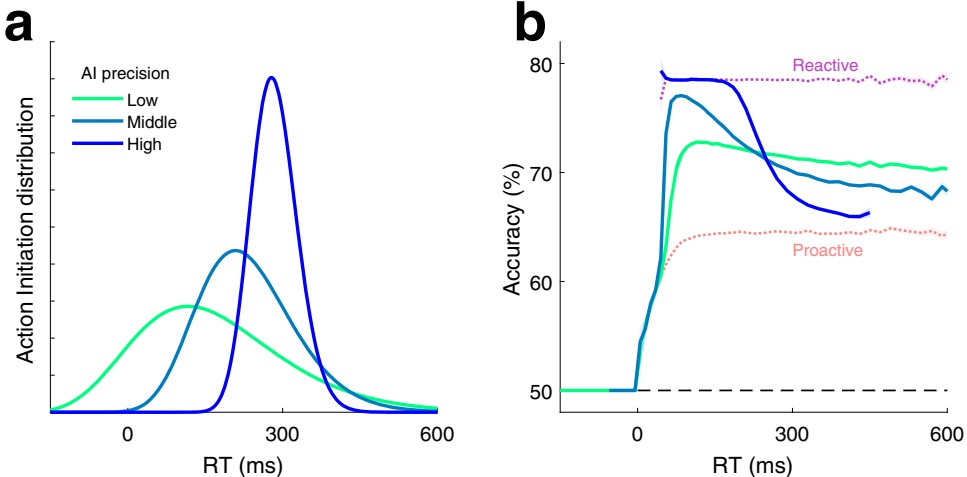

**Fig. 8 Modulation of accuracy with manipulations of AI. a** Distribution of proactive alone RTs for three simulated manipulations varying the latency and the degree of temporal precision of the AI. **b** Simulated tachometric curves obtained from the PSIAM using the AI processes shown in panel a and stimuli of intermediate strength ($s = 0.5$). Contaminant responses were not incorporated into the model. We also simulated tachometric curves for the proactive (pink) and reactive (purple) responses; these two tachometric curves are independent of AI manipulations, and provide lower and upper bounds on accuracy, respectively. Source data are provided as a Source data file.

model separating the *when* and *what* decisions into two separate processes could nicely explain RTs and choices: the Action Initiation process *always* determined the timing of response, whereas a second stimulus-dependent Action Preparation process determined the choice but never the timing[17,18]. While this model, like the PSIAM, can generate proactive responses where the choice is informed by the Evidence Accumulation process, it cannot explain that some reaction times depend on stimulus evidence, a clear signature of reactive responses present in our data (Figs. 3 and 5) and in many other studies[24,40].

An alternative model, based on data from monkeys performing an oculomotor visually-guided task under strong external time pressure, proposes a race between choice-selective Action Initiation processes which are accelerated or decelerated by the presence of a stimulus[16,57,58]. By combining proactive and reactive processes into a single stimulus-dependent race-to-bound process, the model can capture the existence, like in our data, of two underlying components of the RT distribution: responses that are time-locked to the onset of the foreperiod (the fixation onset in our task) and responses that are time-locked to the stimulus onset[16]. The PSIAM differs however from this model in two important aspects. First, the PSIAM incorporates the classical mechanisms of Evidence Accumulation, which captures the dependency of RTs on stimulus strength in standard free-response paradigms, something not present in the accelerated race model of ref. [16]. Second, the PSIAM grounds on separate accumulation-to-bound EA and AI processes, whereas the accelerated race model features proactive and reactive mechanisms in a single accumulation-to-bound process. A single accumulation-to-bound process is incompatible with responses where the stimulus impacts choice but not RT, such as express responses in our rats. Thus, our data support that reactive responses are mediated by a separate accumulation-to-bound process independent from the proactive AI process. As expected, the accelerated race model displayed chance-level performance for responses faster than non-decision time, which contradicts the observed behavior for rats (Supplementary Fig. 7g-h,l and Supplementary Methods). A direct extension of the PSIAM, inspired by the accelerated race model, is to assume that EA does not trigger responses when reaching bound but rather accelerates the ongoing AI process toward the Go bound[16]. This extension of PSIAM would reconcile the two models, and could account for the increased EA latency found in advanced trials (Fig. 5d, g).

Based on the large body of existing mechanistic network models, an implementation of the PSIAM can be readily foreseen. The network would consist of two circuits: a standard circuit carrying out Evidence Accumulation based on inhibition-mediated competition between excitatory populations representing each of the alternatives[59–62]; and a circuit generating stochastic ramping activity which performs the motor timing part of the task[27,63,64]. Coupling between the two circuits could implement the instantaneous collapse of the bounds in the EA circuit when the AI circuit reaches a certain level of activity. Various modulations of the EA circuit may affect the speed of the decision dynamics, such as stronger external feedforward excitation[51,60,65–67], increased top-down modulatory inputs[61,68], a change in the balance between recurrent excitation and inhibition[69] or the impact of different neuromodulators[70,71]. A sudden and large acceleration of the winner-take-all dynamics through either of these mechanisms could terminate the accumulation and categorize the evidence accumulated so far: for example, an all-or-none population burst generated by the AI circuit upon threshold-crossing[72] could generate a strong and fast boost to the competition that would cause the population with higher firing to rapidly increase its rate until reaching the decision threshold. Hence, the rate of the winning population would always be reaching the same decision threshold, as consistently found across multiple brain areas[4,40,73–77], in both proactive and reactive responses.

Where in the brain could these circuits be located? Activity in several brain areas shows correlates of processes like the AI or EA of the PSIAM. Neurons showing slow ramping activity preceding proactive responses have been found in several brain areas such as the cortical supplementary motor area[12], the frontal eye field[16], the secondary motor cortex[11], the lateral intraparietal area[78], and the basal ganglia[13,79]. Interestingly, some neurons showed ramping activity specifically before proactive but not reactive responses[12,78]. Similarly, a large number of cortical and subcortical areas has been suggested to encode accumulated stimulus evidence[4,40,73–77]. There is conflicting evidence regarding whether neural markers of accumulated evidence are influenced by

the level of urgency[22,77,80,81]. Thus, despite being a phenomenological model aimed to describe behavioral responses, by postulating the existence of race dynamics between proactive and reactive processes, the PSIAM model may become an instrumental tool to identify the underlying mechanisms of neural ramping dynamics during perceptual decisions.

In conclusion, our analyses support a model where decision-making is mediated by two separate processes, Action Initiation and Evidence Accumulation, which race to trigger a behavioral response. They also point to the AI process as a flexible mechanism allowing strategic and idiosyncratic adjustments of the action timing to multiple factors, such as temporal and choice predictability, motivation, and urgency.

## Methods

This study complies with all relevant ethical regulations for animal testing and research. All experimental procedures were approved by the local ethics committee (Comité d'Experimentació Animal, Universitat de Barcelona, Spain, Ref 390/14).

**Animal subjects**. Animals were male Long-Evans rats ($n = 26$, 350–650 g; Charles River), pair-housed during behavioral training and kept on stable conditions of temperature (23 °C) and humidity (60%) with a constant light-dark cycle (12 h:12 h, experiments were conducted during the light phase). Rats had free access to food, but water was restricted to behavioral sessions. Free access to water was provided on days with no experimental sessions.

**Task description**. Rats in Group 1 ($n = 10$) performed an auditory reaction-time two-alternative forced-choice task. Specifically, this group of rats performed a frequency discrimination version of the task[23]. Briefly, at each trial, an LED on the center port indicated that the rat could start the trial by poking in (Fig. 1). After a fixation period of 300 ms, the LED switched off and an acoustic stimulus consisting of a superposition of two amplitude-modulated frequencies was presented (see details below). Each frequency was associated with a specific side and reward was available at one of the two lateral ports, depending on the dominant frequency. Stimulus sequences were structured in trial blocks with either a Repeating tendency (prob. repeat previous stimulus category = 0.7–0.8) or an Alternating tendency (prob. repeat = 0.2) (see Supp. Methods). Animals developed expectation-mediated choice biases reflecting this structure but only after correct trials (biased trials). After error trials (unbiased trials), their choices were not influenced by the block[23]. Animals could respond any time after stimulus onset. Correct responses were rewarded with a 24 μL drop of water and incorrect responses were punished with a bright light and a 5 s time-out. Trials in which the rat did not make a side poke response within 4 s after leaving the center port were considered invalid trials and were excluded from the analysis (average of 0.4% invalid trials per animal). Withdrawal from the center port before stimulus onset canceled stimulus presentation (Fixation Break, FB). After a FB, rats were allowed to initiate fixation again, with no time-out, and as many times as necessary until fixation was complete (indicated by center LED offset). Rats performed an average of 694 trials per session (range: 335–1188), one session per day lasting 60–90 min, 6 days per week, during 9 months. The data presented in this study was taken from the period after training yielding an average of 56,506 valid trials per rat.

Rats in Group 2 ($n = 6$), Group 3 ($n = 8$), and Group 4 ($n = 4$) performed an intensity level discrimination version of the task[23,24]. In this task, two speakers positioned at both sides of the box played simultaneously an amplitude-modulated white noise. Rats had to discriminate the side with the loudest sound (right or left) and seek reward in the associated port. The details of the task are exactly the same as in the previous one, but in these groups the time-out after error responses was of 2 s and was not associated with bright light. Rats in Group 2 performed sessions with silent catch trials (random 10% of trials without sounds; range: 45–50 consecutive sessions) in which they could respond any time after the 300 ms fixation offset, as in standard trials. The correct response side for silent trials was determined by the stimulus category sequence, which was predetermined irrespective of the presence or absence of sound (see Supplementary Methods). Rats in Groups 2 and 3 performed sessions with advanced/delayed stimulus catch trials (random 10% of trials varying sound onset; 45–50 consecutive sessions). For 5% randomly selected trials, the stimulus onset time was advanced by $\Delta = -150$ ms for animals in Group 2 and by $\Delta = -250$ ms for rats in Group 3 (Fig. 5a, advanced stimulus trials); and for another 5% of trials, it was delayed by $\Delta = +150$ ms for Group 2 and by $\Delta = +50$ ms for Group 3 (Fig. 5a, delayed stimulus trials). For these advanced/delayed catch trials, stimulus strength was always set to maximum ($s = 1$) to gain statistical power, and to enhance evidence-triggered, reactive responses. In advanced stimulus and standard trials, rats could respond any time after fixation offset (either at 150 or at 50 ms, and at 300 ms, respectively), which coincided with stimulus onset. In delayed stimulus trials, rats could respond any time after fixation offset (300 ms), which was dissociated from stimulus onset (either at 450 or at 350 ms). For rats in Group 2, which participated in silent and

advanced/delayed stimulus sessions, the order of the sessions was balanced across animals, with 3 rats starting with the advanced/delayed catch trials sessions and 3 starting in the silent catch trials sessions. The overall fraction of catch trials was maintained low at 10% to avoid animals modifying substantially their original behavior during these sessions.

Rats in Group 4 performed sessions with a modified water reward size. In small-reward sessions, the amount of water obtained in each correct response (12 μl) was half of the reward in standard sessions. In large reward size sessions, the amount of water obtained (48 μl) was double the amount in standard sessions (Fig. 7a, left). The three types of sessions were randomly interleaved.

For Group 1, the behavioral setup (Island Motion, NY) was controlled by a custom software developed in Matlab (Mathworks, Natick, MA), based on the open-source BControl framework (http://brodylab.princeton.edu/bcontrol). For Groups 2-4, the behavioral setup was controlled by BPod electronic boards (by Sanworks) and the task was run using the Python-based open software package PyBPod (http://pybpod.com/).

Additional details on the acoustic stimulus and the stimulus sequence are provided in the Supplementary Information Methods.

**Reaction times**. The reaction time (RT) was defined as the time elapsed from stimulus onset to center port withdrawal (Fig. 1b). For standard trials, the RT matched the stimulus duration, as the stimulus was switched off at center port withdrawal. For silent, advanced and delayed stimulus trials, the RT was measured with respect to standard stimulus onset. Fixation breaks (FBs; 16 ± 5% of total withdrawals, mean ± standard deviation) were defined as withdrawals from the center port during the fixation period (300 ms for standard, silent and delayed stimulus trials; either 150 or 50 ms for advanced stimulus trials). Only the first center port withdrawals of each trial, either FB or RT, were analyzed: after a FB, further FBs and the subsequent valid response were discarded to remove possible serial effects within a single trial. Trials with RTs longer than 1s were removed from the analysis (0.5 ± 0.6% of total withdrawals).

**RT-evidence modulation onset**. The onset times for the modulation of the RT distributions by the stimulus strength (Fig. 2a, b) were computed similarly as in Supplementary Fig. 2b of[24]. For each time $t$, we computed a one-tailed Kolmogorov–Smirnoff test comparing the RT cumulative distributions for trials with strongest and weakest stimulus evidence (stimulus strength $s = 1$ versus stimulus strength $s = 0$), including both biased and unbiased trials, and excluding all reaction times larger than $t$. For each rat, we defined the modulation onset as the minimal value of $t$ at which this comparison became significant ($p < 0.05$).

**Time delay curves**. Time delay curves assess how much faster or slower is a given stochastic time variable with respect to a reference distribution; see Supplementary Fig. 3 of ref. [24]. Intuitively, it represents the horizontal distance at any time point between a given cumulative distribution and a reference cumulative distribution (Supplementary Fig. 2). At a given time point $T$, the time delay value is defined as the time difference $T_r - T$, where $T_r$ is the time at which the value of the reference cumulative matches the value of the cumulative of interest at $T$, i.e., $C(T) = C_r(T_r)$. Positive time delay means that, at time $T$, there is a larger proportion of RTs lower or equal to $T$ in the condition of interest than in the reference condition.

**Parallel Sensory Integration and Action Model (PSIAM)**. In order to characterize the patterns of rat RTs and choices, we built the PSIAM by combining a standard DDM with constant bounds[10] (i.e., Evidence Accumulation process or EA process) with a second drift diffusion process modeling the timing of proactive responses (Action Initiation process or AI process; Fig. 3a, b, bottom red traces). EA is a Wiener process which integrates the evidence provided by the stimulus over time, following:

$$\frac{dx(t)}{dt} = V_E + \sigma_E \cdot \xi(t), \text{for } x \in (-\theta_E, +\theta_E) \tag{1}$$

$$\frac{dx(t)}{dt} = 0, \text{for } x = \pm\theta_E$$

where $x(t)$ is the EA process representing the instantaneous accumulated evidence. The parameter $V_E$ is the drift which only depends on the stimulus evidence (see below); $\sigma_E^2$ is the noise variance, set to 1 to make the model identifiable; and $\xi(t)$ is a white noise stochastic process which represents the stimulus temporal fluctuations as well as the internal noise associated with the accumulation process. The EA process is initiated at value $x = z_E$ (starting offset) with a delay of $t_E^{aff}$ with respect to stimulus onset $t_{stim}$ (afferent latency); $x = \pm\theta_E$ are the two symmetric decision bounds (Fig. 3a, b, and see below). The model did not include variability in the starting offset when capturing the distribution of RTs for analytical tractability. To predict choices, however, we included a small initial variability in biased trials to capture the fact that the magnitude of the bias was not fixed across all the trials within a block (i.e., the repeating bias fluctuated in magnitude across trials of the Repeating block; see "Model prediction of choice data" section below).

Action Initiation represents the preparation of a commitment to take a decision; it is modeled as an independent, one-bound DDM[1]:

$$\frac{dy(t)}{dt} = V_A + \sigma_A \cdot \xi(t), \text{ for } y < \theta_A \qquad (2)$$

$$\frac{dy(t)}{dt} = 0, \text{ for } y = \theta_A$$

where the parameter $V_A$ is the drift of the AI; $\sigma_A^2$ is the noise variance (set to 1); and $\xi(t)$ is a white noise stochastic process representing the internal variability associated with the precise timing of the Go action. The AI process is initiated at fixation onset with a delay of $t_E^{aff}$ (afferent latency); and $\theta_A$ is the action bound (Fig. 3a, b).

The PSIAM model distinguishes two types of responses, proactive and reactive, depending on the outcome of the race between the AI and the EA processes, i.e., depending on which process reaches the bound first. Reactive reactions are generated when either of the two EA boundaries is reached first; the choice is then defined by which bound is hit (by convention, the upper bound corresponds to the rightwards choice, and the lower bound to the leftwards choice). Proactive responses are generated when the AI threshold is reached first; the choice is then defined as a direct read-out of the sign of the EA process $x(t)$ after the interrupted stimulus is integrated. This is equivalent to an instantaneous collapse of both EA decision thresholds, i.e., a vertical bound in the EA (Fig. 3a). The RT at each trial is thus set by the first of the two processes to reach the bound, and corresponds to the time elapsed from stimulus onset to first bound hit, with an additional efferent delay $t_{motor}$ representing motor latency. The probability density function (pdf) $p(t)$ and the cumulative distribution function (cdf) $c(t)$ of the RTs depend on the pdf and cdf of the RTs that would be generated by either the AI ($p_A(t)$ and $c_A(t)$) or by the EA ($p_E(t)$ and $c_E(t)$) in isolation, through[82]:

$$p = p_A \cdot (1 - c_E) + p_E \cdot (1 - c_A) \qquad (3)$$

$$c = c_A + c_E - c_A \cdot c_E \qquad (4)$$

where we have dropped the dependence on $t$ to ease the notation. The distribution of the RTs generated in isolation by the AI is an inverse Gaussian (IG) distribution (a.k.a. Wald distribution), so that the pdf and cdf for AI[1,83] are:

$$p_A(t|V_A, \theta_A, t_A) = \theta_A \cdot \frac{1}{\sqrt{2\pi \cdot (t - t_A)^3}} \cdot \exp\left(-\frac{V_A^2}{2} \cdot \frac{[(t - t_A) - \theta_A/V_A]^2}{(t - t_A)}\right) \qquad (5)$$

$$c_A(t|V_A, \theta_A, t_A) = \Phi\left(V_A \cdot \frac{(t - t_A) - \theta_A/V_A}{\sqrt{t - t_A}}\right) + \exp(-2 \cdot V_A \cdot \theta_A) \cdot \Phi\left(-V_A \cdot \frac{(t - t_A) + \theta_A/V_A}{\sqrt{t - t_A}}\right) \qquad (6)$$

where the parameter $V_A$ is the reduced drift ($\sigma_A = 1$) of the AI; $\theta_A$ is the reduced single-threshold height from the starting value of the AI process; and $t_A = t_A^{aff} + t_{motor}$ is the total AI latency, summing afferent and motor latencies. The independent RT pdf for the EA alone corresponds to that of the standard DDM with two constant bounds[1,84]:

$$p_E(t|V_E, \theta_E, Z_E, t_E) = p_E^+ + p_E^- = p_E^-(t| - V_E, \theta_E, -Z_E, t_E) + p_E^-(t|V_E, \theta_E, Z_E, t_E) \qquad (7)$$

$$p_E^-(t|V_E, \theta_E, Z_E, t_E) = \frac{\pi}{(2\theta_E)^2} \cdot \exp\left(-V_E \cdot (Z_E + \theta_E) - \frac{V_E^2}{2} \cdot (t - t_E)\right) \cdot \sum_{k=1}^{\infty}\left[k \cdot \sin\left\{k \cdot \pi \cdot \frac{Z_E + \theta_E}{2\theta_E}\right\} \cdot \exp\left(-\frac{k^2\pi^2}{2 \cdot (2\theta_E)^2} \cdot (t - t_E)\right)\right] \qquad (8)$$

where $p_E^\pm$ are the joint pdfs for RT and choice, with $+$ and $-$ for choice corresponding to upper bound and lower bound, respectively. The parameter $V_E$ represents the reduced drift ($\sigma_E = 1$); $\pm\theta_E$ represent the reduced upper and lower threshold values; $Z_E$ represents the reduced starting offset or bias; and $t_E = t_E^{aff} + t_{motor}$ represents the total response latency time for the EA, summing afferent and motor latencies. The corresponding cumulative distribution $c_E$ is obtained by integrating Eq. (7) analytically over $t$.

**Model fit to reaction times data**. The fit of the PSIAM to the experimental RTs data was performed under maximum likelihood estimation (MLE), using simplex method (Matlab function *fmincon*) with 20 random initial points in bounded parameter space to avoid local minima. Model parameters were estimated for each animal only based on RTs for all trials including FBs. The value of the EA starting point, EA drift and AI drift depended on trial conditions. EA drift $V_E$ scaled linearly with signed stimulus strength $S$ (positive/negative sign for rightward/leftward evidence), and we fitted the proportionality parameter $v_E$, i.e., $V_E = v_E \cdot S$. For a given trial $k$, the signed stimulus strength $S_k$ was computed as the instantaneous evidence $S_{k,f}$ averaged across frames presented in this trial, i.e., $\langle S_{k,f}\rangle_f$. To account for the slowing of the responses within each session (Fig. 6a, b), AI drift scaled linearly with trial index $k$ as $V_{A,k} = v_{A0} + v_{trial} \cdot k$, and we fitted the parameters $v_{A0}$ and $v_{trial}$. EA initial offset $Z_E$ captured the animal's expectation of which response

will be rewarded in the next trial. For unbiased trials, this offset was set to 0, i.e., no expectation. For biased trials (Supplementary Figs. 3 and 5), the value of the starting point was signed depending on the trial-dependent expectation of rewarded side $b_k = \pm 1$ (see Supplementary Methods), so that $Z_E = z_E \cdot b_k$, where we fitted the parameter $z_E$ representing the magnitude of the animal's expectation. The rest of the parameters of the PSIAM were constant for each animal. Additionally, we included contaminants in the form of a Huber's ε-contamination model for MLE robustness[85]. Contaminant responses were included to avoid estimation parameters to be biased by very early FBs (especially numerous after error trials) and very late RTs[86]. Contaminant responses were modeled from the mixture of a decaying exponential distribution and a uniform distribution as:

$$p = c \cdot p_C + (1 - c) \cdot p_{model} \qquad (9)$$

$$p_C(t|d, \beta) = d \cdot \beta \cdot \exp(-\beta \cdot t) + (1 - d) \cdot ctt \qquad (10)$$

where $p$, $p_C$, and $p_{model}$ are the final, contaminant, and PSIAM densities respectively; the parameter $c$ is the proportion of contaminant responses; $d$ is the exponential-uniform mixture parameter; $\beta$ is the inverse of the exponential time constant; and ctt is the normalization constant for the uniform density alone. The PSIAM with contaminants has a total of 11 parameters: 4 for the AI (drift intercept $v_{A0}$ and trial-index weight $v_{trial}$, threshold $\theta_A$, and response latency time $t_A$); 4 for the EA (drift stimulus weight $v_E$, threshold $\theta_E$, response latency time $t_E$, and starting offset $z_E$); and 3 parameters for the contaminant distribution (proportion of contaminant responses $c$, the exponential-uniform mixture parameter $d$, and the inverse of the exponential time constant $\beta$). We added lower and upper bounds in the parameter search, both to speed up the optimization algorithm and to confine the search within a range of a priori defined values. The bounds in parameter space set for MLE were: $[0,12]$ s$^{-1}$ for $v_{A0}$; $[-2,1] \times 10^{-2}$ s$^{-1} \cdot$trial$^{-1}$ for $v_{trial}$; $[0.1,10]$ for $\theta_A$; $[-600,300]$ ms for $t_A$; $[2,10]$ s$^{-1}$ for $v_E$; $[0.1,1.2]$ for $\theta_E$; $[35,75]$ ms for $t_E$; $[-1/2,1/3]$ for $z_E$; $[0,0.5]$ for $c$; $[0,1]$ for $d$; and $[0,50]$ s$^{-1}$ for $\beta$. There was one less parameter to fit for unbiased trials as we set $z_E$ to 0. The infinite series in the EA independent RT pdf (Eq. 8) and cdf were approximated numerically[87–89].

**Model prediction for choice data**. The PSIAM choice (Right/Left) corresponds to the sign of the final value of the EA process (Fig. 3a, b). For each rat, we computed the choice prediction (Figs. 3e and 4d) by simulating the PSIAM (Eqs. 1–2) using MLE parameter values. It should be stressed that parameter MLE values were obtained using RTs data only, and not choices. We also included a small trial-to-trial variability in the EA starting offset when predicting choices in biased trials only (see Supplemental Methods). Had we not included this initial variability, choices at very short RT would always be determined by the side of $Z_E$, i.e., the expectation, without any choice variability (i.e., in a Repeating block, the model would always repeat its previous choice). This is at odds with rats' behavior which at very short RTs shows choice variability independently of their expectation bias. For unbiased trials after error responses, because $Z_E = 0$, there was no need to include variability in the EA starting offset to generate choice variability at any RT. Simulations were run by discretizing differential equations (Eqs. 1–2) using Euler method with a time step of 0.1 ms. We performed a total of $10^7$ trial simulations per rat.

**Model prediction for reaction times in catch trials with manipulated stimuli**. The density $p_{silent}$ of RTs in silent catch trials was predicted from the PSIAM using parameters fitted to the standard trials from the same sessions (Fig. 5b; Supplementary Figs. 11, left column and 12a, blue lines). Since no stimulus was presented in silent trials, the prediction was that $p_{silent} = p_A$ (Eq. 5), because responses were all proactive, i.e., were generated from the AI process operating in isolation (AI alone in Fig. 5b; red lines in Supplementary Fig. 11 and blue lines in Supplementary Fig. 12b).

For advanced/delayed trials, where the stimulus onset was advanced or delayed by $\Delta$, we first tested the prediction obtained from simply advancing or delaying the EA onset time $t_{stim}$ by $\Delta$. In a second analysis we fitted the value of the latency time $t_E$ and/or EA drift $v_E$ separately for each delay condition, leaving other parameters unchanged (Fig. 5d, e, g, h; Supplementary Figs. 13 and 14, dashed blue curves, and 15).

**Extended drift-diffusion model**. We constructed an extended version of the classic DDM where the integration of evidence started before stimulus onset, in order to account for very fast reaction times and fixation breaks[28]. The eDDM is initiated at fixation onset with an afferent delay $t_I^{aff}$ and integrates internal noise of variance $\sigma_I^2$ with drift zero prior to stimulus onset. Following stimulus onset with an afferent delay $t_E^{aff}$, eDDM is driven by stimulus evidence: the drift changes to $V_E$ while noise variance changes to $\sigma_S^2$ as both external and internal sources contribute to the accumulation noise. The integration continues until the decision variable reaches a bound at $\pm\theta$. The choice is then defined by which bound is hit, and the response is initiated after an efferent delay $t_{motor}$ representing motor latency. Additionally, we incorporate the same structure of contaminant responses as in PSIAM (see Eqs. 9 and 10) to avoid biases in parameter estimation and improve model comparison.

The model was fit to individual rat RTs in unbiased trials using the PyDDM package[90], under the method of differential evolution. We fitted a total of 8 parameters: pre-stimulus internal noise variance $\sigma_I^2$ and response latency $t_{latency} = t_I^{aff} + t_{motor}$, drift stimulus weight $v_E$, non-decision time $t_E = t_E^{aff} + t_{motor}$, threshold $\theta$, proportion of contaminants $c$, contaminant exponential-uniform mixture parameter $d$, and the inverse of the exponential time constant $\beta$. $\sigma_S^2$ is set to 1 to make the model identifiable.

**Reporting summary**. Further information on research design is available in the Nature Research Reporting Summary linked to this article.

## Data availability

All the data generated in this study have been deposited in the Open Science Foundation database. Data and Source data are publicly available at https://osf.io/3qe59/. Source data are provided with this paper.

## Code availability

All custom code and software in this study (Matlab R2019a, Spyder: Python 3.7) have been deposited in the Open Science Foundation database, and are publicly available at https://osf.io/3qe59/.

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

## Acknowledgements

We thank Jordi Pastor for excellent discussions and Lejla Bektic and Lorena Jiménez for help with training of the animals. We thank Dani Linares, Klaus Wimmer, Anke Braun, Juan R. Castiñeiras, and Alfonso Renart for helpful comments on an earlier version of the manuscript. This research was supported by the Spanish Ministry of Economy and Competitiveness together with the European Regional Development Fund (BES-2011-049131 to A.H.-M.; IJCI-2016-29358 to D.D.; SAF2015-70324-R and RTI2018-099750-B-I00 to J.R.; PSI2015-74644-JIN and RYC-2017-23231 to A.H.), the European Research Council (ERC) under the European Union's Horizon 2020 research and innovation program (grant agreement No. 683209—PRIORS to J.R.). Part of this work was developed at the building Centre Esther Koplowitz, Barcelona.

## Author contributions

A.H.-M. and J.R. designed the first set of experiments (Group 1); A.H.-M. carried out the first set of experiments; L.H.-N., D.D., A.H., and J.R. designed the second set of experiments (Groups 2–4); D.D. carried out the second set of experiments; L.H.-N. analyzed the data; L.H.-N. and A.H. developed the generative model; all authors interpreted the data; L.H.-N., A.H., and J.R. wrote the manuscript with contributions from the rest of the authors.

## Competing interests

The authors declare no competing interests.
