## [Peer Review File · Nature Communications]

A race between Proactive and Reactive processes during perceptual decisionsREVIEWER COMMENTS

Reviewer #1 (Remarks to the Author):

In this interesting paper, Hernández-Navarro et al. proposed a novel variant of the drift-diffusion model (DMM) and tested their model in experimental data. The authors showed that, in addition to the standard evidence accumulation process with the fixed decision boundary (EI), there needs to be a separate stochastic boundary collapse (AI). I found the paper interesting, but I'm a little confused about how the model is justified in the experimental data. I hope the authors can clarify my confusion.

1) Long RT vs. accuracy and lapse responses. The authors showed the subjects' lower choice accuracy at longer RT trials as evidence for the AI process. But can't this instead be explained by the lapse responses? The standard EI + lapse (with no AI) model would not predict this?

2) I also suggest better describing the lapse response of the full model in the main text. It's currently hidden and very confusing.

2) Silent trials. It's nice that the full model can predict RT in silent trials. But what would the standard DDM (EI) predict in these trials? I guess it's just a random walk with no drift, so the RTs would look like the data? Also, I may have missed it. But it would be nice if the authors could clarify what animals (should) do in silent trials. Do they make any choice?

3) Advance and delayed stimulus trials. The authors showed that when stimulus onsets are delayed, there is a second bump in the RT distribution. But the second bumps are very hard to see in most subjects and never quantified. Could the author statistically quantify the existence of the second bumps across subjects?

4) Lack of formal model comparison against other models. If I didn't misunderstand, the authors never formally tested their model (AI + EI + lapse) against other classic models such as EI + lapse, e.g., using BIC. One concern is that the full model has many parameters (table 15) -- so it may be overfitting. Is the AI process BIC-justifiable?

Reviewer #2 (Remarks to the Author):

Nature Communication 284115-0 review

This is a well-written article, describing interesting novel analyses and modelling on data from rats performing auditory discrimination tasks. The article addresses the mechanisms of anticipatory responses and their relationship with stimulus-informed responses. This is an important question that certainly deserves scrutiny and is of interest across the field of decision in human and animal behaviour. The authors propose a new model to account for their data, whereby an action initiation (AI) process is responsible for fast anticipatory responses, while an independent evidence accumulation (EA) process explains how accuracy increases with reaction time. The model is interesting and apparently captures the data very well. Overall, the methods and results are well described, the figures are clear and informative. I have a few concerns regarding the methods and results (points 1 and 2 below), but my main unease is with the rationale for the modelling choices, how they are introduced and discussed in relation to existing modelling work, and how the results are interpreted.

1) Methods. It wasn't clear to me whether the EA process was equipped with trial-to-trial variability in the starting point when fitting the RT data. ZE is included, but as a bias, while its variability doesn't appear as a free parameter in the "Model fit to reaction times data" section. This was only mentioned in the "Model prediction for choice data", and described as having a beta distribution, which is not typical in the DDM framework. Considering that trial-to-trial variability in starting point was specifically added to the drift diffusion model (DDM) to produce fast responses with low accuracy, this seems particularly relevant here. Relatedly, it would be interesting to see what the model fit would have been would the EA alone (with starting point variability) had been fitted to the data, rather than in conjunction with the AI process, for comparison.

2) Results. One of the main predictions of the independent AI system is that anticipatory responses would show the exact same RT with and without the presence of a stimulus, with the latter only affecting choice. The "silent trials" condition is used to show that this is the case. However, the way the data is shown doesn't allow the reader to see this important result with their own eyes, as all we can see are two pictures side by side on one example rats (the other rats are in supplementary info). It would be more informative to compare the model fit of the AI parameters with and without a stimulus, and to statistically test for the evidence against a difference between these across all rats. Relatedly, model fits are sometimes presented on different figures or panels compared to observed data (eg. figure 2 and 3), it would be clearer to show them on top of each other, so one can get a grasp of the goodness of fit.

3) Introduction and rationale for the modelling choices. I do not a priori object that the model may be a sensible or even the only choice, but I would like to see this more thoroughly discussed. Currently, the independent AI process is presented as the unique solution to capture the data, while addressing a limitation of the DDM framework, portrayed in a way that appears simplified to me and presented as the "standard" model. Only when reaching the Discussion are previous modelling attempts at addressing similar phenomena introduced. Although I appreciate that some readers may need to be introduced the basic principles of the DDM as developed in 78, I feel that more recent attempts should be explicitly referred to when explaining the present modelling choice.

The issue posed by very fast responses (be they stimulus-driven or anticipatory or more likely a by-product of pro-active adjustments and very fast sensory signals) has been acknowledged a while ago and solutions have been proposed to improve models' ability to capture these. Importantly, this is not to say that these attempts would necessarily capture the present data well, and maybe there is indeed a need to invent a new model with two separate decision processes whereby one process reaching the boundary creates an immediate boundary collapse in the other one. But the authors should consider whether alternatives, including simpler ones, exist.

To my eyes, the idea that the only thing that can be accumulated in a DDM is evidence related to an external event, as portrayed in the introduction, is a simplification. It is true that the DDM and other accumulator models have been used to capture perceptual decisions where what is accumulated is directly linked to a stimulus, but this is one application of the framework, rather than "inherently" all they can do, or a "key assumption". The DDM is accumulating inputs, but it is not pre-determined what these inputs are about, or whether they are internal or external. It has been used in all sorts of contexts where endogenous signals or noise can influence choice and RT, including value-based decision making and pro-active biases. Indeed, trial-to-trial variability of the starting point can reflect variable pro-active adjustments within a single decision process. The possibility to accumulate time-varying endogenous signals as well as internal noise from an extreme starting point means that the DDM, and other models, could produce anticipations at chance level, if their parameters allowed for it. One reason why they wouldn't often do so is that human behaviour doesn't need that much of it. But to my eyes, it is inaccurate to say that they "invariably predict a coupling between RT and stimulus strength for all responses".

The proposed model is one interesting fix to the DDM's incapacity to offer a biologically plausible mechanism for the effect of non-specific signals (affecting both response options similarly), be they urgency signals or pro-active signals like is the topic here. Tempering with the bounds is a fix that has been proposed in previous work, as mentioned in the Discussion: bringing both bounds closer to each other is equivalent to what would be captured as a non-discriminate boost of activity towards all response options in an accumulator framework. The issue is not fundamentally different between urgency and anticipatory signals, it is therefore unclear to me why the modelling option of having very low bounds at the start of the trials was not considered or even discussed as a viable alternative to having two independent accumulation processes.

The DDM is presented as standard but, although it is indeed quite popular in the field, race accumulator models provide prominent (and maybe more biologically plausible) alternatives. For instance, I am not sure why the authors did not attempt to use the model introduced by Stanford et al. 2010 (described in Discussion), with the very minor upgrade of making the rise rate dependent on the strength of the evidence.

More complex and less common modelling choices could also have been referred to, such as the Trappenberg JoCN 2011 neural field model of eye-movements. This model accounts for express saccades, as well as pro-active adjustments (see Bompas et al. Psychological Rev 2020) by integrating inputs from different sources, exogenous and endogenous, within a single integrator. Based on this principle, a noisy accumulator model could easily capture anticipations as well as stimulus-driven responses, and generate bimodal RT distributions. I appreciate these models are less easy to use and can be perceived as less standard / classical, but I feel their principles could be usefully discussed, especially when these were set partly to describe the “possible coexistence of proactive and reactive processes [and] the mechanisms describing their possible interactions”.

To summarise the few points above, the very existence of possible alternatives to the independent AI process justifies a more careful wording of the results interpretation: I am not yet convinced that the data demonstrate the existence of such an independent AI process. More generally, I feel the introduction needs careful re-writing, avoiding the simplification and doing better justice to the field. It would also be interesting to discuss the likely neural implementation (how would reaching a bound into the pro-active decision system would create a boundary collapse into the other one).

Minor points

- why not start with negative RT on all tachometric curves, since you get plenty of these? It would be nice to see all curves nicely converge to 50% and would allow to compare the time course of accuracy rise between silent trials and normal trials
- The fatigue and satiety analyses are nice, but it isn't clear to me that the effect of reward size reverses, as is claimed throughout this paragraph. It is clear that fast responses are more numerous and decrease quicker in the large reward condition, but the three fitted lines converge. Or is the linear fit not the right one?
- “The existence of the dual mechanism for response timing in the PSIAM suggests that the speed of response could be regulated by two different mechanisms”. Having designed a model in a certain way doesn't “suggest” any mechanisms. It would be more accurate to say: “The existence of the dual mechanism for response timing in the PSIAM offers two possible mechanisms to account for changes in speed of response”.
- “The second basic hypothesis of the PSIAM is that proactive responses are triggered by a process time-locked to fixation onset”. Would it make sense to think it is rather time-locked to the time when the stimulus is meant to onset, which typically coincides with fixation offset, but not always? Within an accumulator framework, one could envisage a ramping up of baseline activity, with a peak at the expected stimulus onset. This would account for the effect observed on advanced trials.

- “We assessed whether the systematic slowing of the RTs observed within each session in every animal (Fig. 6a) was mediated by the modulation of AI, at odds with the classical account of slowing based on decision bounds”. I am not sure what is this “classical” account the authors are referring to. To my knowledge, changes in speed have been accounted for by changes in each and every parameter in all simple accumulator and DDM models, including bounds, but also drift rate, starting point, non-decision time, and their respective variability.

- “They are usually considered as random anticipated responses and removed from subsequent analysis”. Usually should be replaced by sometimes. Express saccades have been the topic of a lot of research. The authors cite Carpenter and Williams 1995, but it is noteworthy that the same author soon showed that these were not random at all (see for instance Carpenter 2001 Vision Research). Many have shown express saccades can be more accurate than chance and that their RT can depend on stimulus properties (Bompas & Sumner 2009 J Neurophysiology is one example that comes to my mind, although I am sure the authors can find more recent accounts). Depending on the experimental conditions such as gap or overlap, express saccades will show a continuum from anticipations at chance level, to above-chance responses later.

- Van den Brink et al., no date. Milosavljevic et al., no date. Shouldn't these be cited as archive paper and given the year they were added online?

- “Furthermore, our model-based approach allowed us to identify the cognitive mechanisms underlying this effect” should say “one possible cognitive mechanism”

- Are contaminant responses (methods) the same as lapses (results)?

Aline Bompas (I sign all my reviews)

We thank the reviewers for their numerous and valuable comments which we think have helped us to significantly improve the quality of the results and the clarity of the presentation. Below we respond to each comment, providing a detailed response together with the extracts from the manuscript that we have added or modified in order to address them.

Our answers to the reviewers are marked in green, and manuscript text is cited in black or blue for unchanged and changed text, respectively. Page numbers refer to the manuscript with tracked changes.

Source data and code will be made available after publication. Reviewers can access the files via the anonymous link:

https://osf.io/3qe59/?view_only=c72e9155becb44e098d758fdd9a31f7e

Reviewer #1 (Remarks to the Author):

In this interesting paper, Hernández-Navarro et al. proposed a novel variant of the drift-diffusion model (DDM) and tested their model in experimental data. The authors showed that, in addition to the standard evidence accumulation process with the fixed decision boundary (EI), there needs to be a separate stochastic boundary collapse (AI). I found the paper interesting, but I'm a little confused about how the model is justified in the experimental data. I hope the authors can clarify my confusion.

1.1) Long RT vs. accuracy and lapse responses. The authors showed the subjects' lower choice accuracy at longer RT trials as evidence for the AI process. But can't this instead be explained by the lapse responses? The standard EI + lapse (with no AI) model would not predict this?

We thank the reviewer for raising this important point. First, we should stress that our main piece of evidence in favor of the PSIAM (and which rules out DDM and its extensions) comes from behavior at short rather than long reaction times. Still, as rightfully pointed out, lapses (renamed as “contaminants”, see point 1.2 below) can cause a decay in accuracy at long RTs in the absence of the AI. We have now fitted an extended DDM (eDDM) with contaminants to the data (see point 4 below), and show that the tachometric curves for such eDDM also produced decay for long RTs (Fig. 4d dotted lines; copied below). However, this decrease in accuracy for long RTs of the eDDM+contaminants is not as strong as in the PSIAM because in the PSIAM both proactive responses and contaminant responses contribute to the decay.

In the original submission we did not put too much attention on contaminant responses to make things simpler and to emphasize the role of the proactive responses at long RTs. We now see from the reviewer's comment that this decision has introduced ambiguity about the role of proactive responses and contaminants in producing the decay in tachometric curves. We have therefore clarified this point in the Results, where we have added the fitted DDM+contaminants for comparison (p. 9):

Moreover, because in those trials the evidence level is lower than the decision bound, their associated accuracy should also be lower than for reactive choices with the

same RT. Contaminant responses, included in the model as rare responses that occur at an approximately constant rate independently of the AI or EA processes, also increase in relative frequency at long RTs (black lines in Fig. 4a-b; Supplementary Figs. 8 and 9; see Methods); and because they yield random choices, their occurrence also predicts lower accuracy. Hence, the increase in the fraction of both proactive and contaminant responses should thus be accompanied by a reduction in categorization accuracy. We tested this prediction by examining tachometric curves at these very long, infrequent RTs and found a significant decrease in accuracy for long RTs (Fig. 4c-d). Although contaminants alone could also produce a decay of the tachometric curves at long RTs, proactive responses in the PSIAM contributed significantly to accentuate and advance this decrease (Fig. 4d). In sum, AI did not only promote express responses but it also shaped slow responses, hindering their occurrence at the expense of reduced accuracy for long RTs.

Selected panels of Figure 4. **c**, Tachometric curves for different stimulus strength s values, median across rats. Open circles indicate accuracy for responses of RT > 600 ms. Shaded areas and error bars: median absolute deviation. **d**, Simulated PSIAM tachometric curves up to 700 ms, median across rats; shaded areas: median absolute deviation. Dotted lines: simulated eDDM curves.

Moreover, we have modified the abstract to qualify the role of proactive AI-responses at long RTs as “contributing” instead of “explaining” urgency as before:

“The Action Initiation process readily explains premature responses, contributes to urgency effects at long reaction times and mediates the slowing of the responses as animals get satiated and tired during sessions.”

Finally, we would like to remind the reviewer that despite the fact that in our data contaminants also contribute to the decay in accuracy for long RTs, that does not imply that proactive responses may not be the only source of decay in the TCs in other experiments (e.g. with primates) in which lapse responses (i.e. contaminants) play a marginal role. This is shown in Fig. 8, in which we simulated data using the contaminant-free PSIAM with different AI precision: as shown in Fig. 8b, depending on the parameters of the AI, the decay of the tachometric curves varies widely in amplitude. In sum, we think that our data provides

evidence of a significant contribution of proactive responses in the decay of the TC and that more generally, proactive responses represent a novel mechanism that can act as an urgency signal preventing very long RTs.

(1.2) I also suggest better describing the lapse response of the full model in the main text. It's currently hidden and very confusing.

We thank the reviewer for noting this confusion. Actually we referred to lapse responses as “contaminant responses” in the Methods. We now use “contaminants” consistently throughout the manuscript. We also now introduce these contaminant responses in the main text (see cited text above in the response to point 1.1).

2) Silent trials. It's nice that the full model can predict RT in silent trials. But what would the standard DDM (EI) predict in these trials? I guess it's just a random walk with no drift, so the RTs would look like the data?

We thank the reviewer for raising this point. We have now fitted an extended DDM with contaminants and integration of noise with no drift prior to stimulus onset. This extended DDM can generate, like the rats, anticipatory responses with very short RTs (or even fixation breaks). However, in contrast with the data and the PSIAM, these anticipatory responses are random (i.e. stimulus-independent) since they are triggered by noise fluctuations (Fig. 3f,h and Supp. Fig. 7j, copied in point 4 below). For silent trials, as for the PSIAM, we predicted the silent RT distributions in each rat from the extended DDM based on parameter fits to RTs in standard trials. Unlike the PSIAM, the distributions of predicted silent RTs did not match the distributions observed experimentally (Supplementary Fig. 12, copied below). The better prediction of the PSIAM was quantified in each rat using the Log-Likelihood of predicted silent trials. We have added the following to the revised Results section of the manuscript (p. 11):

“Crucially, the distribution of RTs in silent trials was accurately predicted by the distribution of proactive responses alone obtained from fitting the full PSIAM in standard trials (Fig. 5c; Supplementary Figs. 11 and 12b). By contrast, the extended DDM failed to capture the distribution of RTs in silent trials ($\Delta LLH < -25$ for all rats; Supplementary Fig. 12b; see Supplementary Methods).”

For the referee's convenience, we paste here the new panels that we added to show this new analysis:

Supplementary Figure 12. PSIAM and eDDM in silent sessions. **a**, Experimental RT distribution (gray bars), eDDM fit (black line) and PSIAM fit (blue line) for maximum stimulus strength ($s=1$) in unbiased standard trials taken from sessions with silent catch trials for rat Group 2 (rows 1-6, numeric label is rat index). Vertical dashed line: stimulus onset. Last row: BIC difference between eDDM and PSIAM. **b**, Experimental RT distributions (gray bars), eDDM predictions (black line; Methods) and PSIAM predictions (blue line; Methods) for unbiased silent trials for the same rats shown in **a**. Last row: Negative log-likelihood model difference.

Also, I may have missed it. But it would be nice if the authors could clarify what animals (should) do in silent trials. Do they make any choice?

Thank you for pointing to this, this was not explicit in the manuscript. In silent trials, as for standard trials, the rewarded response side is based on the predetermined reward sequence. Because this sequence is partially predictable in Repeating and Alternating blocks (see Methods), rats can guess the rewarded side above chance performance in silent trials. We have now added two sentences in the main text (p. 10-11):

“(In silent trials) we omitted the presentation of the stimulus in a subset of trials but still rewarded one of the two responses.”

“In silent trials, animals made choices using internal estimates instead of the stimuli.”

Furthermore, we now detail in the Methods section (p. 23):

“Correct response side for silent trials was determined by the stimulus category sequence, which was predetermined irrespective of the presence or absence of sound (see Supplementary Methods).”

3) Advance and delayed stimulus trials. The authors showed that when stimulus onsets are delayed, there is a second bump in the RT distribution. But the second bumps are very hard to see in most subjects and never quantified. Could the author statistically quantify the existence of the second bumps across subjects?

The referee correctly signals that the effect of delayed stimulus of $\Delta=+150$ ms is rather modest on RT distributions, mostly because the stimulus is played so late that in most trials the rat has already left the port beforehand. The effect is however much more clearly viewed in time delay curves, a much more sensitive measure that can compare the RT distributions for delayed versus silent trials. These time delay curves represent the best tool we could think of to quantify the existence of the bump. Indeed, the impact of late stimulus presentation is very clear, statistically significant, and well predicted by the PSIAM (see dotted line in Fig. 5f).

In the corresponding section of the main manuscript we now state (p. 11):

“This small impact of delayed stimulus was clearly visualized in the experimental time delay curves which showed that the RTs in delayed stimulus trials were significantly advanced with respect to that of silent trials, as predicted by shifting the time delay curve of standard trials by 150 ms (Fig. 5f; p-value = 0.007 for delayed vs silent time delay curves at RT = 300 ms, one-sided t-test, Group 2).

4) Lack of formal model comparison against other models. If I didn't misunderstand, the authors never formally tested their model (AI + EI + lapse) against other classic models such as EI + lapse, e.g., using BIC. One concern is that the full model has many parameters (table 15) -- so it may be overfitting. Is the AI process BIC-justifiable?

We have now fitted to each animal RT data an extended version of the DDM that features integration prior to stimulus onset as well as contaminant responses. The model can indeed produce anticipatory responses, although formal model comparison using BIC did show that the PSIAM captured RT distributions far better without overfitting ($\Delta\text{BIC} < -100$ for all rats); hence the AI process is very much BIC-justifiable.

Our main piece of evidence to rule our classic models is however qualitative, rather than quantitative: the DDM and its extensions, and any model relying on a single boundary mechanism for response initiation and choice selection, fail to produce stimulus-dependent

choices with RTs lower than the non-decision time, i.e. before stimulus starts modulating RTs. We failed to fully convey this point in the previous version of the manuscript. We agree with the reviewer that a formal model comparison should add weight to this conceptual point, and thus we have now clarified the argument and developed it further with new model simulations, model comparison and additional figures.

Besides the extended DDM, we simulated the accelerated race-to-threshold model of Stanford & Salinas, whereby Left and Right choice-selective Action Initiation processes are accelerated or decelerated by the presence of a stimulus (Nature Neuroscience 2010). Moreover, we comment on the DDM with changing bounds. As introduced in the paragraph above, the failure of all these models to capture rat experimental data is related to the fact that response initiation and choice selection rely on the same boundary crossing mechanism, and thus if the stimulus influences response timing it will also influence choice selection, which can only happen after the non-decision time (typically >50 ms). So these models are fundamentally incapable of explaining what we observed in rat “express responses”: very fast (<50 ms) stimulus-informed choices (Fig. 2c-d) despite stimulus independence of RTs (Fig.2a-b).

For instance, the extended DDM and the accelerated race-to-threshold of Stanford and Salinas (Nature Neuroscience 2010) produced chance responses for very short responses (RT<50 ms), whereas rats performed above chance even for RTs < 5 ms. This major discrepancy leads us to conclude that these very fast stimulus-informed responses cannot be produced by boundary crossing in the stimulus evidence space, and therefore points towards the decoupled mechanisms of the PSIAM.

In Results sections and in Fig. 3f-h and Supp Fig. 7, we introduce these two alternative models (p. 7):

“The existence of stimulus-informed express responses is incompatible with a decision mechanism where the timing of the response relies only on the accumulated evidence reaching a decision boundary. Although such a mechanism can generate noise-triggered express responses, choices are only influenced by the stimulus for RTs larger than the non-decision time, which is typically at least 50 ms. We confirmed this by fitting an extended DDM (eDDM), featuring integration of internal noise prior to stimulus onset (Laming 1979), to individual rat RTs. Pre-stimulus noise integration produced anticipatory responses as those seen in rats, although a formal model comparison showed that the PSIAM captured the overall RT distribution much better ($\Delta\text{BIC} > 100$ for any rat in Groups 1 and 2, $n=16$; Supplementary Figs. 7j-k and 12a; see Supp. Methods). Crucially, the eDDM consistently predicted chance performance for express responses (<50 ms, Fig. 3f-h, Supplementary Fig. 7f), unlike what we observed in animals. The limitation to generate informed choices earlier than sensorimotor delays is general to the joint RT-and-choice decision mechanism: it also applies if the DDM bounds change in time (Drugowitsch et al. 2012) or in the accelerated race-to-threshold model (Stanford et al. 2010), where the stimulus modulates a race-to-threshold between anticipatory signals (Supplementary Fig. 7g-h,i; see Supp. Methods). In conclusion, the influence of stimulus on choice in express responses was very well captured by the PSIAM but was fundamentally at

odds with standard models of decision-making relying on a single boundary mechanism for response initiation and choice selection.”

We paste here the new panels that we added showing this new analysis for the referee’s convenience:

Selected panels of Figure 3. **f**, Accuracy on express response trials (RTs shorter than 50 ms) as a function of stimulus strength s for rats (gray), and for model predictions from the PSIAM (blue) and the extended DDM (black, eDDM), both fitted to individual rat RTs. Lines: average across rats (Group 1, $n = 10$). Error bars: s.e.m. **g**, Simulated time delay curves, generated from the eDDM fit to RTs for each animal, median across rats. Shaded areas: median absolute deviation. **h**, Simulated tachometric curves, predicted from fitting the eDDM to each animal’s RTs, median across rats. RT bin width is 10 ms; shaded areas: median absolute deviation.

Supplementary Figure 7. Model comparison. **a-f**, Legend as in Figs. 2b,d and 3d-e,g-h of the main manuscript, respectively. Lines: median across rats (Group 1, $n = 10$). Shaded areas: median absolute deviation. **g-h**, Legend as in Fig. 3g-h of the main manuscript for an example simulation of

the accelerated race model (Supplementary Methods). Lines: mean across realizations; s.e.m. smaller than line width. **i**, Temporal onset of the modulation of rat choices by stimulus, versus estimated non-decision time for PSIAM (blue tilted crosses) and eDDM (black solid circles). Each symbol represents a simulated animal. Choice modulation onset was defined as the shortest time at which the accuracy difference between strongest and weakest stimulus trials (stim. strengths $s = 1$ and 0 , respectively) became significant ($p < 0.01$, two-sided paired t-test, 5-ms time bins). Choice modulation onset for experimental data is presented on the right (green crosses). **j**, RT distribution for an example rat (gray bars; rat #14) and model fits for PSIAM (blue line) and eDDM (black line) for stimulus strength $s=0.25$. Vertical dashed line denotes stimulus onset. **k**, Distribution of differences in Bayesian information criterion between eDDM and PSIAM for rat Group 1. Positive values indicate that the PSIAM provides a better account of experimental RT distribution. **l**, Simulated example RT distribution for the accelerated race model.

We have also expanded our discussion of these phenomenons in the Discussion (p. 19 and 21):

“Extensions to the DDM where the integration process starts prior to stimulus onset can produce anticipatory responses (Supplementary Fig. 7j). However, they inherently predict that if the RT is faster than the non-decision time, and thus independent of the stimulus evidence, then choice should also be independent of the stimuli (Fig. 3f-h and Supplementary Fig. 7e-f).”

“The PSIAM captured rat RT distributions better than an extended DDM (Supplementary Figs. 9j-k and 12a), and it parsimoniously predicted rat RTs in silent trials with remarkable accuracy (Fig. 5c and Supplementary Figs. 11 and 12b).”

“As expected, the accelerated race model displayed chance-level performance for responses faster than non-decision time, which contradicts the observed behavior for rats (Supplementary Fig. 7g-h,l and Supplementary Methods).”

Finally, note that a new paper by Hawkins and Heathcote has presented a model very similar to the PSIAM with a race between a timing process (identical to our AI mechanism) and an evidence integration process (similar to our EA process). The study shows that the model provided a better account of human behavior than DDMs in a variety of decision-making tasks. Despite very strong similarities, our two articles were developed in total independence. We believe that this convergence between different species provided more compelling evidence overall in favor of parallel proactive and reactive processes in the mammal brain. We now comment on this in the Discussion section (p. 19):

“Indeed, a model very similar to PSIAM (featuring a race between a timing process and an evidence accumulation process) was recently and independently proposed, providing a better account of human behavior than generalized DDMs in a variety of decision-making tasks (Hawkins and Heathcote 2021). This convergence of findings between species provided compelling evidence overall in favor of parallel proactive and reactive processes in the mammal brain.”

Reviewer #2 (Remarks to the Author):

Nature Communication 284115-0 review

This is a well-written article, describing interesting novel analyses and modelling on data from rats performing auditory discrimination tasks. The article addresses the mechanisms of anticipatory responses and their relationship with stimulus-informed responses. This is an important question that certainly deserves scrutiny and is of interest across the field of decision in human and animal behaviour. The authors propose a new model to account for their data, whereby an action initiation (AI) process is responsible for fast anticipatory responses, while an independent evidence accumulation (EA) process explains how accuracy increases with reaction time. The model is interesting and apparently captures the data very well. Overall, the methods and results are well described, the figures are clear and informative. I have a few concerns regarding the methods and results (points 1 and 2 below), but my main unease is with the rationale for the modelling choices, how they are introduced and discussed in relation to existing modelling work, and how the results are interpreted.

1.1) Methods. It wasn't clear to me whether the EA process was equipped with trial-to-trial variability in the starting point when fitting the RT data. ZE is included, but as a bias, while its variability doesn't appear as a free parameter in the "Model fit to reaction times data" section. This was only mentioned in the "Model prediction for choice data", and described as having a beta distribution, which is not typical in the DDM framework. Considering that trial-to-trial variability in starting point was specifically added to the drift diffusion model (DDM) to produce fast responses with low accuracy, this seems particularly relevant here.

We thank the reviewer for raising this point. We wanted to include the minimal set of mechanisms required to capture the individual pattern of choices and RTs. We thus did not include variability in the starting point when fitting the model to RTs, as it would have made the fitting procedure much more complex and it was not necessary to generate fast RTs: the AI mechanism was precisely introduced to capture those express responses. As explained below in the response to the referee's question 3.1, adding starting point variability at stimulus onset to the DDM could not capture express responses nearly as well as the AI mechanism proposed in the PSIAM.

A small starting variability in EA, however, was required to capture the animals' *choices* for very early RTs, but only for biased trials (shown in Supp. Fig. 6 only). We now realize that the description of this point was very concise and could lead to misinterpretation, and thus we now expand the explanation about the starting point variability for biased trials (p. 28):

We also included a small trial-to-trial variability in the EA starting offset when predicting choices in biased trials only (see Supplemental Methods). Had we not included this initial variability, choices at very short RT would always be determined by the side of Z_E , i.e. the expectation, without any choice variability (i.e. in a Repeating block, the model would always repeat its previous choice). This is at odds with rats' behavior which at very short RTs shows choice variability independently of their expectation bias. For unbiased trials after error responses, because $Z_E=0$, there was no need to include variability in the EA starting offset to generate choice variability at any RT.

Because simulations of PSIAM choices for biased trials are only presented in Supplementary Figures, we have provided the detailed explanation about the starting point variability in the Supplementary Methods:

For PSIAM simulations of biased trials, we included a small trial-to-trial variability in the EA starting offset (otherwise choices for very short RTs in biased trials would always be determined by the side of Z_E , i.e. the expectation, unlike what is observed in rats). The distribution of starting point was thus taken as a Beta distribution (stretched to cover the full domain of the EA process $[-\theta_E, \theta_E]$). The parameters of the distribution were set such that it had mean z_E and standard deviation equal to 10% of the interbound distance between bounds (i.e. $0.2 \cdot \theta_E$).

The beta distribution is a typical choice of distribution for a bounded variable (as the initial value of the starting point is bounded between the two decision boundaries). For the range of parameters we chose, with a high concentration around the mode, it closely resembles the truncated gaussian distribution, which is the other typical distribution for a bounded variable, as shown in the figure below.

(1.2) Relatedly, it would be interesting to see what the model fit would have been would the EA alone (with starting point variability) had been fitted to the data, rather than in conjunction with the AI process, for comparison.

We thank the reviewer for the suggestion. In the previous version of the manuscript, there was a conceptual point about why existing models (including the DDM with starting point variability) cannot capture experimental data and which we failed to fully convey. We have now further clarified and developed the argument, as well as included a formal model comparison with additional figures in the new version of the manuscript.

The failure of previous models to capture rat experimental data is *not* a failure to produce anticipatory responses. It is related to the fact that, because in these models response initiation and choice selection rely on a single boundary crossing mechanism, then if the stimulus influences response timing it will also influence choice selection, which can only happen after the non-decision time (typically >50 ms). So these models are fundamentally incapable of explaining what we observed in rat “express responses”: very fast (<50 ms) stimulus-informed choices (Fig. 2c-d) despite stimulus independence of RTs (Fig.2a-b).

To clarify this point, and following the reviewer's suggestion, we have fitted to each animal RT data an extended version of the DDM that features integration prior to stimulus onset (which generates variability in the starting offset at stimulus onset), as well as contaminant responses. The model can indeed produce anticipatory responses, although formal model comparison using BIC did show that the PSIAM captured RT distributions far better ($\Delta\text{BIC} < -100$ for all rats). Crucially, the DDM produced chance responses for very short RTs (<50 ms), whereas rats performed above chance even for RTs < 5 ms (compare in Fig. 3 panels e and h). This major discrepancy leads us to conclude that these very fast stimulus-informed responses cannot be produced by boundary crossing in the stimulus evidence space, and so points towards the decoupled mechanisms of the PSIAM. These results are now described in Figure 3f-h and Supplementary Fig. 7i-k, copied below.

Here is our report of these new additions in the revised manuscript. Overall, we strongly believe that these new analyses provide a much stronger case about the necessity for a new model to capture stimulus dependent choices with RTs faster than the non-decision time.

In the Results section we now add (p. 7):

“The existence of stimulus-informed express responses is incompatible with a decision mechanism where the timing of the response relies only on the accumulated evidence reaching a decision boundary. Although such a mechanism can generate noise-triggered express responses, choices are only influenced by the stimulus for RTs larger than the non-decision time, which is typically at least 50 ms. We confirmed this by fitting an extended DDM (eDDM), featuring integration of internal noise prior to stimulus onset (Laming 1979), to individual rat RTs. Pre-stimulus noise integration produced anticipatory responses as those seen in rats, although a formal model comparison showed that the PSIAM captured the overall RT distribution much better ($\Delta\text{BIC} > 100$ for any rat in Groups 1 and 2, $n=16$; Supplementary Figs. 7j-k and 12a; see Supp. Methods). Crucially, the eDDM consistently predicted chance performance for express responses (<50 ms, Fig. 3f-h, Supplementary Fig. 7f), unlike what we observed in animals. The limitation to generate informed choices earlier than sensorimotor delays is general to the single RT-and-choice decision mechanism: it also applies if the DDM bounds change in time (Drugowitsch et al. 2012) or in the accelerated race-to-threshold model (Stanford et al. 2010), where the stimulus modulates a bounded race between anticipatory signals (Supplementary Fig. 7g-h,i; see Supp. Methods). In conclusion, the influence of stimulus on choice in express responses was very well captured by the PSIAM but was fundamentally at odds with standard models of decision-making relying on a single boundary mechanism for response initiation and choice selection.”

We paste here the new panels that we added showing this new analysis for the referee's convenience:

Selected panels of Figure 3. **f**, Accuracy on express response trials (RTs shorter than 50 ms) as a function of stimulus strength s for rats (gray), and for model predictions from the PSIAM (blue) and the extended DDM (black, eDDM), both fitted to individual rat RTs. Lines: average across rats (Group 1, $n = 10$). Error bars: s.e.m. **g**, Simulated time delay curves, generated from the eDDM fit to RTs for each animal, median across rats. Shaded areas: median absolute deviation. **h**, Simulated tachometric curves, predicted from fitting the eDDM to each animal's RTs, median across rats. RT bin width is 10 ms; shaded areas: median absolute deviation.

Selected panels of Supplementary Figure 7. **i**, Temporal onset of the modulation of rat choices by stimulus, versus estimated non-decision time for PSIAM (blue tilted crosses) and eDDM (black solid circles). Each symbol represents a simulated animal. Choice modulation onset was defined as the shortest time at which the accuracy difference between strongest and weakest stimulus trials (stim. strengths $s = 1$ and 0 , respectively) became significant ($p < 0.01$, two-sided paired t-test, 5-ms time bins). Choice modulation onset for experimental data is presented on the right (green crosses). **j**, RT distribution for an example rat (gray bars; rat #14) and model fits for PSIAM (blue line) and eDDM (black line) for stimulus strength $s = 0.25$. Vertical dashed line denotes stimulus onset. **k**, Distribution of differences in Bayesian information criterion between eDDM and PSIAM for rat Group 1. Positive values indicate that the PSIAM provides a better account of experimental RT distribution.

We have also expanded the discussion on the inability of the DDM to capture these responses in the Discussion (p. 19):

“Extensions to the DDM where the integration process starts prior to stimulus onset can produce anticipatory responses (Supplementary Fig. 7j), but they inherently predict that if the RT is faster than the non-decision time, and thus independent of the stimulus evidence, then choice should also be independent of the stimuli (Fig. 3f-h and Supplementary Fig. 7e-f).”

“The PSIAM captured rat RT distributions better than an extended DDM (Supplementary Figs. 9j-k and 12a), and it parsimoniously predicted rat RTs in silent trials with remarkable accuracy (Fig. 5c and Supplementary Figs. 11 and 12b).”

Finally, note that a new paper by Hawkins and Heathcote has presented a model very similar to the PSIAM with a race between a timing process (identical to our AI mechanism) and an evidence integration process (similar to our EA process). The study shows that the model provided a better account of human behavior than DDMs in a variety of decision-making tasks. Despite very strong similarities, our two articles were developed in total independence. We believe that this convergence between different species provided more compelling evidence overall in favor of parallel proactive and reactive processes in the mammal brain. We now comment on this in the Discussion section (p. 19):

“Indeed, a model very similar to PSIAM (featuring a race between a timing process and an evidence accumulation process) was recently and independently proposed, providing a better account of human behavior than generalized DDMs in a variety of decision-making tasks (Hawkins and Heathcote 2021). This convergence of findings between species provided compelling evidence overall in favor of parallel proactive and reactive processes in the mammal brain.”

2.1) Results. One of the main predictions of the independent AI system is that anticipatory responses would show the exact same RT with and without the presence of a stimulus, with the latter only affecting choice. The “silent trials” condition is used to show that this is the case. However, the way the data is shown doesn’t allow the reader to see this important result with their own eyes, as all we can see are two pictures side by side on one example rats (the other rats are in supplementary info). It would be more informative to compare the model fit of the AI parameters with and without a stimulus, and to statistically test for the evidence against a difference between these across all rats.

We thank the reviewer for appreciating the importance of this result. We are not sure however, what is meant by comparing “two pictures side by side”. In our view, the relevant comparison is between the bar plot and the model curve within Figure 5c (and within Supplementary Fig. 11, right column for all rats), which we can see nicely match. As the legend specifies: “RT distribution in silent trials from the same rat shown in b (gray bars). Model prediction corresponds to the RT distribution of proactive alone responses obtained from standard trials (red line; same as in b)”. This shows that fitting the model on the standard conditions allows to fully predict the RT distribution in the silent trials.

Second, we think that this very fine quantitative match between the predicted and observed distribution in silent trials (which is present in all rats; Supplementary Fig. 11, right column) is much more compelling than comparing fitted values of the model. This is because we lose much of the richness of the distribution when we just capture it by a few model parameters, so our comparison is actually much more conservative. To put an example, we can fit two sample distributions with a gaussian and obtain that the mean and standard deviations are equal, but it is more convincing if we can show that the distributions actually match.

Additionally, we have now used fits of the extended DDM in standard trials to predict responses in silent trials and found that the extended DDM could not properly capture the distribution of RTs in silent trials, as opposed to PSIAM. This is now shown in Supplementary Fig. 12, reproduced below. In the Results section we now say (p. 11):

“Crucially, the distribution of RTs in silent trials was accurately predicted by the distribution of proactive responses alone obtained from fitting the full PSIAM in standard trials (Fig. 5c; Supplementary Figs. 11 and 12b). By contrast, the extended DDM failed to capture the distribution of RTs in silent trials ($\Delta LLH < -25$ for all rats; Supplementary Fig. 12b; see Supplementary Methods).”

Supplementary Figure 12. PSIAM and eDDM in silent sessions. a, Experimental RT distribution (gray bars), eDDM fit (black line) and PSIAM fit (blue line) for maximum stimulus strength ($s=1$) in unbiased standard trials taken from sessions with silent catch trials for rat Group 2 (rows 1-6, numeric

label is rat index). Vertical dashed line: stimulus onset. Last row: BIC difference between eDDM and PSIAM. **b**, Experimental RT distributions (gray bars), eDDM predictions (black line; Methods) and PSIAM predictions (blue line; Methods) for unbiased silent trials for the same rats shown in **a**. Last row: Negative log-likelihood model difference.

(2.2) Relatedly, model fits are sometimes presented on different figures or panels compared to observed data (eg. figure 2 and 3), it would be clearer to show them on top of each other, so one can get a grasp of the goodness of fit.

The goodness of fit of the model can be visually assessed in many plots in the manuscript in which we did superpose experimental and simulated distributions of RTs for each individual rat at each stimulus strength (Figs. 3c, 4a, 5b and Supplementary Fig. 7j for example rats and stimulus strengths; Supplementary Figs. 4, 5, and 8 for all rats in Group 1; Supplementary Figs. 11 left column, 12a, 13 center column and 14 center column for all rats in Groups 2 and 3 at maximum stimulus strength $s = 1$). The accuracy versus the RT plots (i.e. tachometric curves) were not fitted by the model but were predicted based on model parameter estimates obtained from the RT distribution. Hence, the quantitative match between the PSIAM and the experimental data regarding accuracy is not completely perfect, but the qualitative match is very good. We have opted to present these tachometric curves for data and model in separate panels, and following the reviewer's comment, we now present them side by side (Supplementary Fig. 7a-h, copied below). If we present them on top of each other, the stress of the figure would be visually put in the quantitative match and not in the qualitative comparison, which shows very convincingly that only the PSIAM captures the data qualitatively, in particular for express responses. Hence, we do not choose this presentation format to hide the quantitative match, but to emphasize a qualitative comparison that all other models fail to capture.

Selected panels of Supplementary Figure 7. Model comparison. **a-b**, Experimental time delay and tachometric curves. Lines: median across rats (Group 1, $n = 10$). Shaded areas: median absolute deviation. **c-f**, Legend as in panels **a-b**, generated from simulations of the PSIAM and eDDM fitted to RTs for each animal. **g-h**, Legend as in panels **a-b** for an example simulation of the accelerated race model (Supplementary Methods). Lines: mean across realizations; s.e.m. smaller than line width.

3.1) Introduction and rationale for the modelling choices. I do not a priori object that the model may be a sensible or even the only choice, but I would like to see this more thoroughly discussed. Currently, the independent AI process is presented as the unique solution to capture the data, while addressing a limitation of the DDM framework, portrayed in a way that appears simplified to me and presented as the “standard” model. Only when reaching the Discussion are previous modelling attempts at addressing similar phenomena introduced. Although I appreciate that some readers may need to be introduced the basic principles of the DDM as developed in 78, I feel that more recent attempts should be explicitly referred to when explaining the present modelling choice.

The issue posed by very fast responses (be they stimulus-driven or anticipatory or more likely a by-product of pro-active adjustments and very fast sensory signals) has been acknowledged a while ago and solutions have been proposed to improve models’ ability to capture these. Importantly, this is not to say that these attempts would necessarily capture the present data well, and maybe there is indeed a need to invent a new model with two separate decision processes whereby one process reaching the boundary creates an immediate boundary collapse in the other one. But the authors should consider whether alternatives, including simpler ones, exist.

To my eyes, the idea that the only thing that can be accumulated in a DDM is evidence related to an external event, as portrayed in the introduction, is a simplification. It is true that the DDM and other accumulator models have been used to capture perceptual decisions where what is accumulated is directly linked to a stimulus, but this is one application of the framework, rather than “inherently” all they can do, or a “key assumption”. The DDM is accumulating inputs, but it is not pre-determined what these inputs are about, or whether they are internal or external. It has been used in all sorts of contexts where endogenous signals or noise can influence choice and RT, including value-based decision making and pro-active biases. Indeed, trial-to-trial variability of the starting point can reflect variable pro-active adjustments within a single decision process. The possibility to accumulate time-varying endogenous signals as well as internal noise from an extreme starting point means that the DDM, and other models, could produce anticipations at chance level, if their parameters allowed for it. One reason why they wouldn’t often do so is that human behaviour doesn’t need that much of it. But to my eyes, it is inaccurate to say that they “invariably predict a coupling between RT and stimulus strength for all responses”.

We thank the reviewer for raising this point. Indeed, the manuscript was not entirely clear as to what exactly the DDM (and extensions) fails to capture in our rat data. We certainly did not want to claim that it is incapable of producing anticipatory choices or random choices, for example by starting the integration before stimulus onset and integrating noise (or internal factors) until reaching the decision. As detailed in the response to point 1, what such a model DOES fail to capture is the fact that choices are impacted by the stimulus even for very short RTs (express responses, faster than the non-decision time of >50 ms typically), while RTs are impacted by the stimulus at a longer latency. This is the core observation that we found in all rats and motivated us to propose the alternative mechanism of responses triggered at a time defined by an external process (the Action Initiation). As detailed in point 1, we have now fitted and simulated a DDM with integration of noise prior to stimulus onset, possibly corresponding to the integration of internal stimulus-unrelated signals, as suggested

by the referee. The model was clearly ruled out by BIC comparison on the RT data, but most fundamentally by its inability to produce above-chance accuracy for express responses. We thank the reviewer as this provides an opportunity to discuss these important extensions to the DDM prior to the Discussion, showing how different modelling approaches capture rat behavior - or fail to.

Finally, we thank the reviewer for pointing out the inaccurate judgment in our discussion, which does not reflect the rationale of why the rat data is incompatible with the DDM and its extensions. We have modified the sentence to (p. 4):

“Express responses are incompatible with standard models of evidence accumulation and their extensions because these models inherently rely on evidence bounding to trigger the response, and therefore they invariably predict **that if the stimulus impacts the choice it should also impact reaction times.**”

(3.2) The proposed model is one interesting fix to the DDM’s incapacity to offer a biologically plausible mechanism for the effect of non-specific signals (affecting both response options similarly), be they urgency signals or pro-active signals like is the topic here. Tempering with the bounds is a fix that has been proposed in previous work, as mentioned in the Discussion: bringing both bounds closer to each other is equivalent to what would be captured as a non-discriminate boost of activity towards all response options in an accumulator framework. The issue is not fundamentally different between urgency and anticipatory signals, it is therefore unclear to me why the modelling option of having very low bounds at the start of the trials was not considered or even discussed as a viable alternative to having two independent accumulation processes.

We thank the reviewer for the suggestion. As discussed in point 1, our rationale is that the DDM is incapable of producing non-random responses for RTs smaller than the non-decision time, irrespective of the precise shape of the boundary. An initial low value of the bounds will indeed generate a larger proportion of rapid responses. However, in all cases the stimulus cannot influence either RT or choice for responses faster than the non-decision time, which captures both sensory processes prior to EA and motor processes posterior to EA. Thus, all responses faster than non-decision time will perform at chance level according to the DDM since this model has a single boundary mechanism for response initiation and choice selection, which is inconsistent with rats’ performance at fast responses (Fig. 3f,h and Supplementary Fig. 7b,f, reproduced in points 1 and 2, respectively).

(3.3) The DDM is presented as standard but, although it is indeed quite popular in the field, race accumulator models provide prominent (and maybe more biologically plausible) alternatives. For instance, I am not sure why the authors did not attempt to use the model introduced by Stanford et al. 2010 (described in Discussion), with the very minor upgrade of making the rise rate dependent on the strength of the evidence.

The accelerated race-to-threshold model suffers from the same limitation as the DDM and its extensions: because the same threshold determines response timing and choices, it cannot produce stimulus-informed choices but stimulus-independent RTs as observed in rat express

responses (also for responses faster than non-decision time). This was previously commented in the Discussion in the section related to this model: “A single accumulation-to-bound process is incompatible with responses where the stimulus impacts choice but not RT, such as express responses in our rats”. Following the referee’s comment, to make the point more explicit, we have now added simulations of the “accelerated race model” with the suggested upgrade (modulation by stimulus strength). As anticipated, the model failed to produce stimulus-informed choices (faster than non-decision time) but stimulus-independent RTs (Supplementary Fig. 9g-h,l, copied below).

Selected panels of Supplementary Figure 7. g-h, Legend as in the selected panels g-h of Fig. 3 for an example simulation of the accelerated race model (Supplementary Methods). Lines: mean across realizations; s.e.m. smaller than line width. **i**, Simulated example RT distribution for the accelerated race model.

We have also included this discussion in the Discussion Section (p. 21):

“As expected, the accelerated race model displayed chance-level performance for responses faster than non-decision time, which contradicts the observed behavior for rats (Supplementary Fig. 7g-h,l and Supplementary Methods).”

(3.4) More complex and less common modelling choices could also have been referred to, such as the Trappenberg JoCN 2011 neural field model of eye-movements. This model accounts for express saccades, as well as pro-active adjustments (see Bompas et al. Psychological Rev 2020) by integrating inputs from different sources, exogenous and endogenous, within a single integrator. Based on this principle, a noisy accumulator model could easily capture anticipations as well as stimulus-driven responses, and generate bimodal RT distributions. I appreciate these models are less easy to use and can be perceived as less standard / classical, but I feel their principles could be usefully discussed, especially when these were set partly to describe the “possible coexistence of proactive and reactive processes [and] the mechanisms describing their possible interactions”.

We thank the referee for pointing out this reference. It is certainly relevant in the context of our paper, particularly because it nicely shows that “endogenous” signals modulate the reaction times in responses driven by exogenous visual inputs. This combination of

exogenous/endogenous factors resembles in some ways our separation of reactive/proactive responses. However, there is a major difference: our proactive process is sufficient to trigger responses independently of the sensory inputs (i.e. of exogenous signals), whereas in the Trappenberg model responses are ultimately triggered by exogenous inputs (endogenous inputs play a modulatory role only).

We have included the following text in the Discussion explaining the similarities and differences between the two models (p. 20):

“The PSIAM is also related to previous models of sensory guided responses with a proactive component. In a classic mechanistic model of saccade initiation, Trappenberg and colleagues presented a circuit model of the superior colliculus where exogenous visual signals and endogenous preparatory signals are combined to generate target-directed saccades with variable RTs (Trappenberg et al. 2001). The model elegantly reproduces the impact of distractors and target location biases on saccade RTs, and generates express saccades when the target onset is cued (see above). However, despite the parallelism between proactive/reactive and endogenous/exogenous processes, this model exclusively generates reactive responses, whose latency can be modulated by endogenous signals, but which are ultimately triggered by the visual input.”

(3.5) To summarise the few points above, the very existence of possible alternatives to the independent AI process justifies a more careful wording of the results interpretation: I am not yet convinced that the data demonstrate the existence of such an independent AI process. More generally, I feel the introduction needs careful re-writing, avoiding the simplification and doing better justice to the field.

We hope that our new results and clarifications of the manuscripts, probed by the referee's comments, will now convince the referee that the coexistence of stimulus-dependent choices and stimulus-independence RTs can only be accounted for by postulating an independent AI process. Regarding the introduction, there is a fragile trade off between giving enough details about the state of the art to clearly motivate the study and putting too many details which might end up obscuring the question for the reader less familiar with these questions. Because the audience of Nature Communications is very broad, we would prefer to stick with a rather simple (but factual) presentation of the state of the art in the introduction, and introduce more refinements progressively in the Results and Discussion sections, as we do now. Please note that in the introduction we now explicitly refer to “the standard DDM” only, and that the most important works related to urgency mechanisms in the DDM and its extensions (Thura et al, Drugowitsch et al, Stanford and Salinas) are acknowledged in the introduction.

It would also be interesting to discuss the likely neural implementation (how would reaching a bound into the pro-active decision system would create a boundary collapse into the other one).

This is a very interesting suggestion. After a careful rereading of the literature on mechanistic model of evidence accumulation and timing we now propose a plausible neural implementation that we detail in the Discussion (p. 21):

Based on the large body of existing mechanistic network models, an implementation of the PSIAM can be readily foreseen. The network would consist of two circuits: a standard circuit carrying out Evidence Accumulation based on inhibition-mediated competition between excitatory populations representing each of the alternatives (Wang 2002; Roxin and Ledberg 2008; Wimmer et al. 2015; Prat-Ortega et al., n.d.); and a circuit generating stochastic ramping activity which performs the motor timing part of the task (Durstewitz 2003; P. Simen et al. 2011; Paton and Buonomano 2018). Coupling between the two circuits could implement the instantaneous collapse of the bounds in the EA circuit when the AI circuit reaches a certain level of activity. Various modulations of the EA circuit may affect the speed of the decision dynamics, such as stronger external feedforward excitation (Furman and Wang 2008; Roxin and Ledberg 2008; Standage et al. 2011; Miller and Katz 2013; Murphy, Boonstra, and Nieuwenhuis 2016), increased top-down modulatory inputs (Wimmer et al. 2015; Lo, Wang, and Wang 2015), a change in the balance between recurrent excitation and inhibition (Lam et al., n.d.) or the impact of different neuromodulators (Hu, Zylberberg, and Shea-Brown 2014; Eckhoff, Wong-Lin, and Holmes 2009). A sudden and large acceleration of the winner-take-all dynamics through either of these mechanisms could terminate the accumulation and categorize the evidence accumulated so far: for example, an all-or-none population burst generated by the AI circuit upon threshold-crossing (Lo and Wang 2006) could generate a strong and fast boost to the competition that would cause the population with higher firing to rapidly increase its rate until reaching the decision threshold. Hence, the rate of the winning population would always be reaching the same decision threshold, as consistently found across multiple brain areas (Roitman and Shadlen 2002; Kim and Shadlen 1999; Yartsev et al. 2018; Joshua I. Gold and Shadlen 2007; T. D. Hanks et al. 2015; Ding and Gold 2013; Thura and Cisek 2014), in both proactive and reactive responses.

Minor points

- why not start with negative RT on all tachometric curves, since you get plenty of these? It would be nice to see all curves nicely converge to 50% and would allow to compare the time course of accuracy rise between silent trials and normal trials.

Unfortunately, we cannot compute tachometric curves for negative RTs because we cannot infer rats' choices for negative RTs. Negative RTs correspond to fixation breaks, which animals can detect since the LED did not turn off when they departed the central port before the end of the fixation period. Upon breaking fixation, the rats almost always repoked immediately in the center port to start a new fixation, without poking in any of the response ports.

- The fatigue and satiety analyses are nice, but it isn't clear to me that the effect of reward size reverses, as is claimed throughout this paragraph. It is clear that fast responses

are more numerous and decrease quicker in the large reward condition, but the three fitted lines converge. Or is the linear fit not the right one?

We thank the reviewer for making this point. We agree that the evidence for a reversal is not convincing. It is in any case tangential to our main point: the slopes are all negative (because of fatigue), but steeper for larger reward size (because of satiety). We have now rewritten this section to avoid mentioning a putative reversal (p. 15):

- “the effect switched at the end of the session” -> “the effect vanished at the end of the session”
 - “the gradual switch of the reward size dependence along the session” -> “the gradual decay of the reward size dependence along the session”
 - “at the end of the session the effect reversed because they were also more satiated” -> “at the end of the session the effect vanished because they were also more satiated”
- “The existence of the dual mechanism for response timing in the PSIAM suggests that the speed of response could be regulated by two different mechanisms”. Having designed a model in a certain way doesn’t “suggest” any mechanisms. It would be more accurate to say: “The existence of the dual mechanism for response timing in the PSIAM offers two possible mechanisms to account for changes in speed of response”.

Agreed - this has been corrected as suggested, thank you for drawing our attention to this point.

- “The second basic hypothesis of the PSIAM is that proactive responses are triggered by a process time-locked to fixation onset”. Would it make sense to think it is rather time-locked to the time when the stimulus is meant to onset, which typically coincides with fixation offset, but not always? Within an accumulator framework, one could envisage a ramping up of baseline activity, with a peak at the expected stimulus onset. This would account for the effect observed on advanced trials.

This is more a terminology debate here. Mechanistically, the AI process starts after fixation onset. The AI can be tuned to reach the threshold after stimulus onset, in this sense it can be said to anticipate stimulus onset, but this is only a temporal prediction. In the related section we precisely study what happens when this prediction fails by shifting the onset time of the stimulus. To make this clearer, we have changed the text to (p. 11):

“The second basic hypothesis of the PSIAM is that proactive responses are triggered by a process initiated with fixation onset, while reactive responses are triggered by a process locked to stimulus onset.”

- “We assessed whether the systematic slowing of the RTs observed within each session in every animal (Fig. 6a) was mediated by the modulation of AI, at odds with the classical account of slowing based on decision bounds”. I am not sure what is this “classical”

account the authors are referring to. To my knowledge, changes in speed have been accounted for by changes in each and every parameter in all simple accumulator and DDM models, including bounds, but also drift rate, starting point, non-decision time, and their respective variability.

We thank the reviewer for this point. A change in the decision bound is considered as the normative approach to adjust the speed-accuracy trade-off (e.g. Gold & Shadlen 2007), but indeed a change in any parameter of the DDM can affect speed. We have changed the end of the sentence to reflect this (p. 13):

“...at odds with the classical account of slowing based on changes in the evidence integration process.”

- “They are usually considered as random anticipated responses and removed from subsequent analysis”. Usually should be replaced by sometimes. Express saccades have been the topic of a lot of research. The authors cite Carpenter and Williams 1995, but it is noteworthy that the same author soon showed that these were not random at all (see for instance Carpenter 2001 Vision Research). Many have shown express saccades can be more accurate than chance and that their RT can depend on stimulus properties (Bompas & Sumner 2009 J Neurophysiology is one example that comes to my mind, although I am sure the authors can find more recent accounts). Depending on the experimental conditions such as gap or overlap, express saccades will show a continuum from anticipations at chance level, to above-chance responses later.

We thank the referee for pointing out that our description of express saccades was inaccurate. From our understanding of the literature on oculomotor tasks, we had established an implicit comparison between the express responses of the rats with the express saccades of the primates based on: (1) express saccades are around 50 ms faster than standard saccades (2) they seem to occur when there is pre-stimulus cueing (e.g. gap paradigm), and (3) they seem to be triggered by a separate circuit or mechanism than the standard saccades (e.g. as suggested by the bimodal distribution of RTs found in some subjects). All these elements are shared with the rats’ express responses and it was for that reason that we established the parallelism. After the referee’s comment, we have revised this comparison and we have realized that it is not clear that these two types of responses are related, and hence we would rather avoid making the comparison. In fact, there are reasons to believe that they are different: express saccades are *reactive responses* because they are triggered by the onset of the target stimulus (Fischer and Weber 1993) with RTs > 70-80 ms, while we precisely define express responses in our rats as those with RT < 85-90 ms, that are faster than any reactive response in our task, and thus they are purely *proactive*. In sum, while we will maintain the name of express responses in the manuscript to describe this fast premature responses of the rats, we now explicitly clarify the possible confusion between the two when presenting express responses in the Results section (p. 4):

(Note that the express responses in our task are not equivalent to express saccades in oculomotor tasks (Carpenter and Williams 1995); see Discussion)

We further make a more detailed comparison of the two response types in the Discussion (p. 17):

The express responses exhibited by our rats should not be confused with the so-called express saccades (R. H. Carpenter and Williams 1995). Express saccades occur at least 80 ms after stimulus onset, around 50 ms before the standard saccades, in conditions when subjects are pre-cued about the onset of the target stimulus (e.g. the gap paradigm) (Carpenter 2001). Despite strong debate about their underlying nature (Fischer and Weber 1993; R. H. S. Carpenter 2001; Trappenberg et al. 2001) it is generally accepted that they are triggered by the onset of the target stimulus (Fischer and Weber 1993) and hence they are reactive responses. The express responses made by our rats are what in psychophysics is called anticipatory responses, i.e. responses which are too fast to be triggered by the stimulus (see e.g. (Laming 1979)).

- Van den Brink et al., no date. Milosavljevic et al., no date. Shouldn't these be cited as archive paper and given the year they were added online?

We thank the reviewer for noting this. The citations have now been corrected.

- "Furthermore, our model-based approach allowed us to identify the cognitive mechanisms underlying this effect" should say "one possible cognitive mechanism"

We have amended the text as suggested.

- Are contaminant responses (methods) the same as lapses (results)?

Indeed. We now refer to "contaminant responses" throughout the manuscript.

Aline Bompas (I sign all my reviews)

REVIEWERS' COMMENTS

Reviewer #1 (Remarks to the Author):

The authors fully addressed my concerns. Thanks very much.

Reviewer #2 (Remarks to the Author):

I thank the authors for their careful and comprehensive revisions and replies. I am now happy with the core of the work and think it offers a very valuable contribution to the field. Below I highlight one main suggestion related to how the work is presented (as well as two minor points).

The peculiarity of their data and why existing models would typically fail to capture them is now clearer to me. I think the reason would deserve a more prominent exposure, although I appreciate this would make the whole paper look more niche.

Currently, one has to wait for the Discussion to see a very important aspect of the data explicitly presented: “because of sensory delays, and because the response involves a full orienting body movement lasting 300-500 ms, the rat had time for finishing stimulus integration between leaving the central port (which interrupts the stimulus) and poking at one of the lateral ports.” Indeed, in the current design, RT and choice are measured at different time, and therefore it is not that surprising that they are, to some extent, decoupled. In practice, the rat may be taking two decisions: one to remove its nose from fixation, and one to poke a peripheral location.

This may be typical in the rodent literature, but is not representative of the field more generally. Most decision models were developed to account for button presses or saccades. Button presses reveal both RT and choice at the exact same time, while saccades are ballistic enough in most designs that RT and choice can be treated as one single event.

Reading the introduction and discussion, the reader is sometimes under the impression that the need for a new model is an indication of the limitations of standard models at capturing choice RT data in general. I would recommend a more careful exposure, that highlights that the need may be specifically for rodent studies because of the specific design they use.

This is a minor point: “Standard models of evidence-to-bound integration predict that, as the stimulus strength increases, accuracy also increases, while reaction time (RT) decreases”. The implication here makes me uncomfortable. First, this is more general knowledge for stimuli strengths within a certain

range than a prediction of any specific model (eg very loud sounds at the pain threshold could lead to longer RT). More importantly, this doesn't mean that the exact quantitative relationship between stimulus strength and rise rate is known, and that any departure is diagnostic of a failure of standard models.

Another minor point: I thank the authors for including mention of the Trappenberg model. However, it is inaccurate to say that this model can only produce stimulus-triggered responses. It could very well produce anticipatory responses, if one sets the baseline endogenous fixation input to a low value. However, these responses would be anticipatory, and driven by noise and at chance level. Therefore, it couldn't account for the observed dissociation here, no more than any model designed to account for decisions that are taken in one step.

Aline Bompas (I sign all my reviews)

Below we respond to each comment, providing a detailed response together with the extracts from the manuscript that we have added or modified in order to address them.

Our answers to the reviewers and editor are marked in green, and manuscript text is cited in black or blue for unchanged and changed text, respectively. Page numbers refer to the manuscript (Word file) with tracked changes.

Source Data and Custom Code and Software are now publicly available at: <https://osf.io/3qe59/>.

Reviewer #1 (Remarks to the Author):

The authors fully addressed my concerns. Thanks very much.

We thank the reviewer for their numerous and valuable comments which we think have helped us to significantly improve the quality of the results and the clarity of the presentation.

Reviewer #2 (Remarks to the Author):

I thank the authors for their careful and comprehensive revisions and replies. I am now happy with the core of the work and think it offers a very valuable contribution to the field. Below I highlight one main suggestion related to how the work is presented (as well as two minor points).

We thank the reviewer for her positive evaluation of our revisions, as well as for her numerous and valuable comments which we think have helped us to significantly improve the quality of the results and the clarity of the presentation.

1) The peculiarity of their data and why existing models would typically fail to capture them is now clearer to me. I think the reason would deserve a more prominent exposure, although I appreciate this would make the whole paper look more niche. Currently, one has to wait for the Discussion to see a very important aspect of the data explicitly presented: “because of sensory delays, and because the response involves a full orienting body movement lasting 300-500 ms, the rat had time for finishing stimulus integration between leaving the central port (which interrupts the stimulus) and poking at one of the lateral ports.” Indeed, in the current design, RT and choice are measured at different time, and therefore it is not that surprising that they are, to some extent, decoupled. In practice, the rat may be taking two decisions: one to remove its nose from fixation, and one to poke a peripheral location. This may be typical in the rodent literature, but is not representative of the field more generally. Most decision models were developed to account for button presses or saccades. Button presses reveal both RT and choice at the exact same time, while saccades are ballistic enough in most designs that RT and choice can be treated as one single event. Reading the introduction and discussion, the reader is sometimes under the impression that the need for a new model is an indication of the limitations of standard models at capturing choice RT data in general. I would recommend a more careful exposure, that highlights that the need may be specifically for rodent studies because of the specific design they use.

We thank the reviewer for raising our attention on this point; still, we disagree with the reviewer's comment. We feel that the standard models have limitations at capturing choice RT data *in general* because they only generate ballistic responses. As explained below, non-ballistic responses that can be updated during their execution are not exclusive to rodent studies. Hence they should be accounted for by models that want to capture the generality of decision making tasks.

We think that the strategy adopted by our animals holds as long as the response trajectory can be modulated or corrected online based on new (sensory) information. This is of course the case in our study where the animal had to reach a choice port from a central port (as in other rodent studies such as Busse et al., *J. of Neurosci.*, 2011; Brunton et al., *Science*, 2013; and Pardo-Vázquez et al., *Nat. Neurosci.*, 2019). But this is also the case in human and non-human primate studies where subjects report their choices by performing a reaching movement with their hand (e.g. Körding & Wolpert, *PNAS*, 2004; Resulaj et al., *Nature*, 2009). Actually this last study investigated how arm movements can be reprogrammed *en route* to a different response point when subjects change their mind after integrating sensory information. We agree that responses through button presses or saccades, widely used in the human literature, probably constitute ballistic movements which allow for marginal updating/modulation. There is however some sign of modulation even in eye saccades (Seideman et al., *Nat. Comm.*, 2018). More importantly, we note that these forms of reporting choices are generally used for experimental convenience. Other forms of responding such as arm or body movements are at least as frequent in natural settings. Thus we would not label our experimental paradigm as *niche* but we think it has in fact a high ecological relevance.

Moreover we are now investigating the details of the orienting trajectories of the animals to understand when the stimulus information is taken into account when animals generate a proactive response. We found no support in our analysis for the idea that rats make their responses in two steps as the reviewer is suggesting. Rather, it seems that for proactive responses rats pre-program a default response and then update the orienting trajectory with the information of the stimulus. This update can occur in less than 100 ms from stimulus onset, which considering the afferent and efferent latencies is a very short time. The total response trajectory takes on average around 300 ms. Why updating an ongoing movement is faster than responding from a still state is a matter of debate (Smeets et al., *Motor Control*, 2016), but our paradigm clearly shows that our animals are able exploit this difference very efficiently generating responses which are both fast and accurate.

We have now modified the Discussion to make this clear, that is to explain that the full trajectory is very fast (300 ms) and that the updating occurs before 90 ms (p. 15):

Second, because of sensory delays, and because the response involves an orienting body movement lasting around 300 ms, the rat had time for finishing stimulus integration between leaving the central port (which interrupts the stimulus) and poking at one of the lateral ports. This makes express responses a viable strategy in which subjects leave the port right after stimulus onset, before stimulus can influence the motor plan, and integrate the sensory information while executing the orienting movement. Preliminary analysis of the orienting trajectories reveals that the

integration of the stimulus information is extremely fast, with the trajectories being updated in less than 90 ms after stimulus onset.

Finally, we note that in a recent and independent study in *Psychological review*, Hawkins & Heathcote (2021) have reanalyzed two human free response perceptual decision-making datasets, with coupled RT and choice timings, showing supportive evidence in favor of a race between proactive and reactive responses over standard models. This suggests that proactive responses may also be used when motor responses are ballistic movements (note that proactive choices in their model did not take stimulus information into account).

Hence, the need for new models is not limited to rodent studies because of their specific design since: (1) similar designs have been extensively used in human perceptual decision-making; and (2) there is evidence that responses are triggered by a race between proactive and reactive responses even in standard designs.

All in all, we agree with the reviewer in that the decoupled timing of RT and choice is a critical aspect of our data, and that it should be further stressed in the Introduction for clarity.

To better convey these points in the revised version of the manuscript, we now include the following changes in the Introduction (p. 2):

Proactive responses are at play in self-paced actions in the absence of stimuli^{11–13}, like in foraging decisions¹⁴, but they can also be particularly prevalent in sensory driven decisions when the stimulus onset can be anticipated¹⁵, or under strong time pressure^{16–19}. Proactive responses could also be a convenient strategy if the response can be updated after it is initiated to incorporate new sensory information.

Minor points

1) This is a minor point: “Standard models of evidence-to-bound integration predict that, as the stimulus strength increases, accuracy also increases, while reaction time (RT) decreases”. The implication here makes me uncomfortable. First, this is more general knowledge for stimuli strengths within a certain range than a prediction of any specific model (eg very loud sounds at the pain threshold could lead to longer RT). More importantly, this doesn’t mean that the exact quantitative relationship between stimulus strength and rise rate is known, and that any departure is diagnostic of a failure of standard models.

We think there was a misunderstanding concerning the meaning of the term “stimulus strength”, which we define in the manuscript shortly before the sentence cited by the reviewer as “The discrimination difficulty of each stimulus (i.e. stimulus strength s) set the relative amplitude of each sound”. We do not mean to refer to a direct physical measure such as loudness, but as the mean net stimulus evidence towards one of the two options. In that sense, we believe that the fact that larger stimulus strength leads to improved accuracy and faster responses admits few exceptions.

2) Another minor point: I thank the authors for including mention of the Trappenberg model. However, it is inaccurate to say that this model can only produce stimulus-triggered responses. It could very well produce anticipatory responses, if one sets the baseline endogenous fixation input to a low value. However, these responses would be anticipatory, and driven by noise and at chance level. Therefore, it couldn't account for the observed dissociation here, no more than any model designed to account for decisions that are taken in one step.

We thank the reviewer for pointing out this inaccuracy in our discussion of the Trappenberg model. We have now amended the text to properly describe this model (p. 18):

However, despite the parallelism between proactive/reactive and endogenous/exogenous processes, this model exclusively generates either reactive responses or proactive random guesses. Reactive responses' latency can indeed be modulated by endogenous signals, but they are ultimately triggered by the visual input.

Aline Bompas (I sign all my reviews)